# Strategic equity compensation: A delphi-AHP approach to industry-specific governance design for RSUs

**Won Albert Park**[1,2], **Elena Sernova**[2], **Cheong-Yeul Park**[1]*

1 Seoul Business School, Seoul, Republic of Korea, 2 Business School Lausanne, Geneva, Switzerland

* cypark@assist.ac.kr

## Abstract

This study views RSUs (Restricted Stock Units) as a strategic tool to achieve sustainable growth, shareholder value enhancement, and key talent retention, and proposes RSUs introduction and operation framework for Korean companies. To this end, a three-round Delphi survey was conducted with 31 experts (11 from the legal and accounting group and 20 from the strategy, HR, and IR group), to derive key decision-making items for each stage of 'strategy setting, execution, evaluation and control'. The panel size fits the appropriate range of existing research, and the content validity and level of agreement were statistically verified using CVR (Content Validity Ratio) and Kendall's W (Kendall's coefficient of concordance (W)). Subsequently, based on the industrial classification system GICS (Global Industry Classification Standard), four industries were classified (consumer goods, resources and energy, industry and infrastructure, technology and communications) and a total of 48 leaders, including C-level, executives, and team leaders, were selected across the four industry groups (12 per group) as panelists and the relative importance and priorities of each item were calculated based on the criterion of a CR(Consistency Ratio below 0.1. The AHP (Analytic Hierarchy Process) results of the legal and accounting groups showed that 'Compliance with relevant standards' and 'Preparation for audit and supervision response' were the top factors, suggesting that the stability of RSUs operations is dependent on regulatory compliance and external supervision response capabilities. In the areas of strategy, HR, and IR, the importance of the strategy execution stage was higher than the strategy setting, evaluation, and control stage in all industries. Consumer goods, technology and communications industries evaluated 'Core talent incentives' and 'Incentive model diversification' as key priorities, while resource/energy and industrial/infrastructure industries evaluated 'RSUs retention period' and 'Short and long-term performance evaluation model' as key priorities. This study contributes by addressing the limitations of previous studies that only derived the relationship between variables such as RSUs introduction or vesting period and scale and financial performance and analyzing RSUs introduction and operation from

**Data availability statement:** All relevant data are within the manuscript and its Supporting Information files.

**Funding:** If this manuscript is accepted for publication, the authors will receive a research grant of 5,000,000 KRW from Seoul Business School (aSSIST University) to support further academic work.

**Competing interests:** he authors have declared that no competing interests exist.

the perspective of the process of 'strategic setting-execution-evaluation and control'. It also demonstrated empirically that a strategy that reflects the characteristics of each industry is needed rather than a uniform introduction of RSUs, provides policy and practical implications for preparing Korea's RSU guidelines, and can serve as a strategic benchmark for countries or companies with similar environments. Unfortunately, this study does not include the financial and healthcare industries, so further research is needed. Additionally, although Delphi-AHP presents priorities, it cannot verify the causal relationship of RSUs on financial performance or shareholder value, so the results of this study can be extended to follow-up verification studies applied to actual data.

## 1. Introduction

There are various compensation systems, but stock options have been evaluated as a key tool for top management's compensation and have been evaluated as a device to strengthen management's motivation. Yet several structural problems with stock options began to surface over time, a point that many early adopters did not anticipate. Stock options have been reported as a key factor in compensation, which tends to make recipients overly obsessed with stock prices in the short term, increasing stock price volatility and encouraging short-termism in core and important decision-making [1]. Especially, there are cases in which management artificially boosts stock prices to realize short-term profits and then exercises stock options, and they are exposed to the problem of damaging the company's competitiveness by using short-term profit maximization strategies rather than long-term and sustainable corporate growth and shareholder value purposes. This is evaluated as a significant risk in that it undermines the sustainability of the company and can lead to a long-term decline in corporate value [2,3].

Restricted Stock Units (RSUs) are drawing attention as an alternative to compensate for the limitations of these stock options and to induce long-term value creation rather than short-term performance [4,5]. RSUs vest only when certain conditions, such as tenure or performance goals over a specified period, are met, and there is no separate exercise price [6]. This structure reduces incentives for RSUs recipients to make excessive investments or accounting adjustments to raise stock prices in the short-term. It also has the advantage of focusing more on mid to long-term performance improvements and corporate qualitative growth [7,8].

In the United States, RSUs have spread in earnest since Microsoft introduced them in 2003 [5], and since the late 2000s, the compensation system has shifted from stock options to RSUs [9,10]. After 2020, empirical results have also been presented that restricted stock compensation structures such as RSUs are more advantageous for improving financial performance [11,12], and RSUs are recognized as a major device that enhances long-term performance and corporate value beyond stock options. Practical research on the RSUs' taxation and vesting mechanism also

emphasizes the importance of RSUs' valuation and tax design [13], suggesting RSUs are establishing themselves as a mature means of compensation at both institutional and practical levels.

In Europe, the introduction of RSUs is relatively slower than in the United States due to strong labor laws, traditional corporate governance, and complex tax systems. Even in this situation, RSUs compensation is gradually spreading among European global conglomerates such as SAP and LVMH [4,5]. In Japan, stock options have long been a major stock-based compensation system like the United States, but in response to changes in the business environment, such as expanding foreign investment and strengthening the outside director system, legal reform for the legalization of RSUs began in earnest from the mid-2010s [14]. Major Japanese companies like Toyota, Sony, and SoftBank have introduced RSUs to compensate for the limitations of stock options, and are now known to use them as a long-term incentive for management and key talent [5,15,16].

Unfortunately, Korea is still in the early stages of introducing the RSUs system, although the situation in major developed countries is like this. Since Hanwha, which is one of the largest Korean conglomerate, first introduced RSUs in 2020, some conglomerates such as Naver, SK, and Doosan have introduced RSUs [17], but there were legal uncertainties and regulatory risks [18] during the introduction process due to the lack of clear legal standards for RSUs, and Doosan, conscious of public opinion due to the misunderstanding [19] that it represents the interests of business owners, sometimes withdrew the introduction of RSUs. Despite this situation in Korea, there is a movement to commercialize RSUs, with the number of companies introducing RSUs doubled from 8 to 16 between 2020 and 2023 [6]. Particularly, in 2025, Samsung Electronics also introduced a restricted stock system based on the nature of RSUs in its executive remuneration system, suggesting that companies are aware of the need for RSUs.

On the other hand, most prior studies focus on verifying ex post performance effects in countries where RSUs are already institutionalized. Research on a systematic decision-making framework for designing the components and operational mechanisms during the initial implementation phase remains severely lacking. Especially in cases like Korea, where characteristics of both developed and emerging economies coexist in terms of governance structures and compensation practices, research is needed on introducing and operating RSUs suited to Korea's institutional environment and industrial characteristics, rather than simple imitation of foreign systems [5,17,18].

This study's main purpose is to establish RSUs introduction and operational framework by major industry sectors that can contribute to the development of RSUs adoption and operational strategies for Korean companies within this context. The methodology employs Delphi–AHP (Analytic Hierarchy Process) multi-criteria decision-making approach.

First of all, through literature research, previous research on stock-based compensation, including RSUs, and various related studies are organized, and an analysis framework for the introduction of the system is established. Next, a Delphi survey is conducted on experts such as strategy, human resource, investor relations, law, and accounting to derive key items necessary for the introduction and operation of RSUs. Based on the items derived afterwards, the relative importance and priorities of each item are derived using the AHP methodology, and a decision-making framework that can systematically check key factors in the process of reviewing the introduction of RSUs by Korean companies and regulators is presented and provided with reference materials for institutional design and operation guidelines. And it is expected that the analysis framework and implications of this study will provide comparison criteria that can be used for the introduction and operation strategy of the RSUs system to companies in emerging countries that have not yet been institutionalized or are in the early stages of introduction, similar to Korea.

Chapter 2 of this paper establishes a theoretical background by reviewing the concept and introduction trend of RSUs and stock-based compensation systems and summarizing previous studies related to behavioral economics perspective and governance, industry-specific heterogeneity, and Delphi–AHP methodology. Chapter 3 explains in detail the research design and analysis method, and details the design, sample composition, model setting, and analysis procedures of Delphi and AHP surveys. Chapter 4 and 5 present the results of Delphi and AHP analysis by industry and analyzes the

priorities of key items and differences between industries. Lastly, Chapter 6 derives the academic and practical implications, discusses limitations, and outlines directions for future research.

## 2. Literature review

### 2.1. Studies on effects of stock compensation

Many quantitative studies have been conducted on the financial performance and governance improvement effects of stock-based compensation such as RSUs. Tai [9] analyzed changes in ROA and Tobin's Q before and after the introduction of RSUs for Taiwanese listed companies, showing that corporate performance significantly improved after the introduction of RSUs. Lian et al. [20] also compared companies that introduced stock incentives and non-introduced companies among Chinese listed companies with propensity score matching and showed that ROA and Tobin's Q significantly increased after the introduction. Further studies have analyzed how performance effects vary depending on contract design variables such as vesting period, grant size, and incentive strength beyond simple introduction. For example, Qiao et al. [21] reported a greater improvement in operational performance since its introduction if the vesting period is long and the grant scale is large.

Research focusing on the informational content of equity compensation costs themselves has also been conducted. Alhaj-Ismail and Adwan [22] found that equity compensation costs for UK listed companies better predict future ROA and ROE than other compensation types. Hundal and Eskola [23] and Aljughaiman et al. [24], analyzing the interaction between CEO compensation structures and governance characteristics, confirmed that equity-based compensation positively impacts ROA and Tobin's Q, but this effect is amplified when the board is independent and has strong oversight functions. Meanwhile, Ma and Wang [25] and Huang and Huang [26] reexamined equity-based compensation, including RSUs, from a human resource management perspective. They demonstrated that equity compensation systems contribute to improved financial performance by mediating through managerial entrepreneurship or human capital efficiency.

Full-scale academic research on RSUs has accelerated since the limitations of the traditional stock-based compensation system centered on stock options began to emerge. Recent studies have commonly argued that RSUs are more effective in increasing long-term performance and shareholder value than stock options.

Lu [27] reported the possibility that RSUs can mitigate stock price volatility and improve financial stability. Bhagat et al. [28] and Hou et al. [12] demonstrated that RSUs induce a transition to long-term profit maximization by reducing incentives to manage short-term stock prices. Murphy and Vance [29] also reported that RSUs promote long-term performance through management's psychological satisfaction and behavioral changes, and Tai [9] analyzed that the introduction of RSUs had a significant effect on sales growth and Tobin's Q, and Lovett et al. [11] argued that RSUs can promote long-term performance creation and strategy execution as a means of compensation suitable for new CEOs. And Murphy and Vance [29] emphasized that RSUs are a strategic compensation mechanism that goes beyond simple financial incentives to change the attitudes and behaviors of management. Qiao et al. [21] showed that the introduction of stock incentive systems and contract characteristics of Chinese listed companies significantly improved operational performance such as ROA and sales growth. Odero [30] presented evidence that the introduction of an employee equity ownership system (ESOP), which is related to equity-based incentives, significantly improves productivity, ROA and EPS.

Studies on stock compensation, including RSUs, do not provide only positive conclusions. Blouin and Carter [31] showed that excessive payment of stock performance compensation, such as RSUs, can limit revenue recognition due to tax constraints and perceived cost burden, and Kanagaretnam et al. [32] pointed out that stock-related compensation systems can weaken the link between compensation and performance when the incentive strength is insufficient. In an empirical analysis of Japanese companies, Hasegawa et al. [14] reported that operating performance tended to deteriorate after the introduction of stock options, and Guo [33] confirmed a relationship in which Tobin's Q significantly decreased when the intensity of executive stock-based compensation increased in a study of Chinese listed companies. These discussions suggest that performance may vary depending on the design, operation, and management of RSUs.

## 2.2. Behavioral finance research on stock-based compensation

Traditional incentive-related contract theory studies have investigated the impact of compensation structure and performance indicator design on top management's decision-making [34,35]. The behavioral corporate finance literature suggests that the same stock-based compensation structure can have contradictory results depending on the behavioral characteristics of managers by combining psychological factors like managerial overconfidence, reference points, and risk perception.

Ben-David and Chinco [36] assumed managers of large publicly traded companies as "EPS (Earnings Per Share) maximizers" and studied how leverage, share buybacks, issuance, and M&A structures change, concluding that EPS-driven compensation and valuation could lead to distortions in business operations. Sloan and Wang [37] used analyst predictions to demonstrate the predictability of EPS growth, showing that such predictable EPS growth is reflected in stock prices and valuation ratios, suggesting a significant portion of valuation comes from expected EPS growth differences.

Studies analyzing restricted stock incentives also exist, premised on managers' attitudes toward risk and psychological reference points. Ma et al. [38] presented findings on S&P 500 firms showing that RSUs do not uniformly reduce CEO risk-taking propensity; instead, they have opposing effects depending on the regulatory focus governing the CEO. Simply, CEOs focused on promotion take on more risk—expanding R&D, capital expenditures, and long-term debt—as the proportion of restricted shares increases. Conversely, prevention-focused CEOs behave more conservatively under the same compensation structure.

Kim and Yang [10] applied a difference-in-differences approach to U.S. listed companies, confirming that after introducing RSUs, both sales and price-to-earnings ratios (PER) increased significantly. They also found that the scale of RSUs issuance consistently improved sales and net income at multiple points before and after issuance (t-1, t, t+1). They interpret this as the expectations and motivation formed among employees during the RSUs vesting period being proactively reflected in performance, and subsequently driving long-term retention and achievement of goals, leading to sustained improvements in financial performance.

Behavioral finance research also focuses on how employees subjectively evaluate stock compensation. Abudy and Benninga [39] estimated that restricted stocks granted to ordinary employees have a discount rate of 12–58% (average about 30%) due to non-marketability and non-diversification, and found that even restricted stocks of the same nominal value may have a different perceived value depending on the risk aversion and portfolio constraints of compensation beneficiaries. Murphy and Vance [29] elaborated that employees exercise stock options early because they need to secure liquidity in real life, and showed that option exercise is focused on large spending points such as housing costs and school expenses, which suggests that liquidity constraints and loss avoidance tendencies should be considered together when designing stock compensation, not from the theoretical perspective explained by financial theory.

The impact of the stock compensation system on the selection of performance indicators and financial reporting behavior is also analyzed from a behavioral perspective. Choi et al. [40] analyzed stock options and RSUs separately and found that RSUs significantly reduced both accruals and actual earnings management to improve the quality of financial reporting, but stock options did not have such an effect. Armstrong et al. [41] showed that a compensation system structure with strong risk incentives, such as stock options, increases the likelihood of distortion of financial reporting and non-disclosure as well.

This aligns with the interpretation that RSUs, with relatively limited downside risk, mitigate profit adjustment incentives compared to stock options and can induce more conservative reporting behavior. Jia [42] analyzing U.S. IT firms and identified a significant positive relationship between restricted stock compensation issuance and ROA, while finding only limited effects on ROE or Tobin's Q. This highlights the divergence between accounting-based and market-based metrics, raising the behavioral choice problem of which metric to set as the core management target when designing restricted stock compensation.

## 2.3. Governance restructuring and stock compensation

After the global financial crisis, regulatory reforms, the rise of shareholder activism, and introduction of stewardship codes propelled executive compensation systems to the forefront of corporate governance restructuring [43]. Empirical studies on the 'say-on-pay' system—where shareholders vote on executive compensation (policy, total amount, structure) decided by the board at shareholder meetings—in the US and UK showed that compensation votes at shareholder meetings help curb excessive pay and partially improve the link between compensation and performance [44,45].

The European Union institutionalized requirements for board compensation committees regarding disclosure of compensation policies, shareholder voting procedures, and long-term incentive structures through the 'Shareholder Rights Directive II' [46]. Comparative law studies synthesizing these regulatory reforms and post-financial crisis compensation debates indicate a trend in major European countries toward gradually increasing the proportion of long-term incentives within equity compensation systems, such as restricted stock and performance-linked stock [47].

By contrast, studies analyzing the relationship between governance reforms and the adoption of equity-based compensation emphasize that the impact of stock-based incentives on performance, risk-seeking behavior, and financial reporting practices varies depending on the industry or institutional context.

Armstrong et al. [41] pointed out the limitations of short-term, centralized incentive design, showing that stock-based compensation with strong risk incentives can be linked to financial reporting distortions for U.S. listed companies. Abudy and Benninga [39] evaluated the value of employee stock options or RSUs in consideration of marketability constraints imposed on unlisted stocks, and pointed out that even rewards of the same nominal value may have different perceived values and incentive effects depending on liquidity and diversification constraints.

In Korea, the stewardship code 'Principle of Trustee Responsibility of Institutional Investors' was announced in 2016 and the adoption of the trustee responsibility principle of the National Pension Service in 2018 strengthened the demand for active shareholder rights exercise by institutional investors and monitoring of the board of directors and compensation committees. And the level of disclosure of management remuneration and shareholder participation are gradually expanding as changes in laws and systems such as mandatory governance reports and improvement of the electronic shareholders' meeting and voting rights proxy exercise system are being promoted [17,48].

Studies regarding RSUs view that these changes in the legal and institutional environment are coupled with discussions on the introduction of stock-based compensation measures, and in recent years, discussions in terms of actual law, taxation, and governance have been active in the fields of law and accounting. Yoon [17] compared Korean and American laws and argued that Korea's demand for RSUs and stock grants is increasing despite relatively strict stock option regulations compared to the United States, and that the legal nature of these compensation systems and measures to prevent damage to shareholder value should be clarified. Kim & Yang [10] and Lee [19] argued that legislative improvements including beneficiary qualifications, disclosure obligations, and tax incentives are needed because the uncertainty of the regulation and taxation system for RSUs in Korea is a key issue limiting the activation of RSUs. In the field of startups and ventures, Hwangbo and Yang [49] used Korean KOSDAQ and unlisted venture companies to point out that RSUs can be an effective means of securing and maintaining key talents in early companies with limited cash compensation capacity, but various regulations such as dividend profit requirements and treasury stock acquisition are major barriers to introducing the system.

## 2.4. Trends of RSUs adoption and industry heterogeneity

Kim and Yang [10] investigated the issuance trend of RSUs and stock options with executive compensation data from 1992–2023 of S&P-listed companies in the United States. The financial ratio was analyzed as a dependent variable for the effect of introducing RSUs additionally. Looking at the changes in 'Table 1' [10] below, it can be interpreted that stock compensation of S&P-listed companies is shifting from stock options to RSUs. The RSUs introduction rate has risen rapidly

**Table 1. RSUs and stock option adoption status of S&P-listed companies.**

| Year | Number of Companies Issuing RSUs (A) | Number of Companies Issuing Stock Options (B) | Number of Companies Reporting Executive Compensation Data to Compustat (C) | Ratio (A/C) | Ratio (B/C) |
|---|---|---|---|---|---|
| 2003 | 723 | 1664 | 1937 | 0.37 | 0.86 |
| 2004 | 885 | 1543 | 1879 | 0.47 | 0.82 |
| 2005 | 958 | 1359 | 1763 | 0.54 | 0.77 |
| 2006 | 1331 | 1306 | 1931 | 0.69 | 0.68 |
| 2007 | 1719 | 1539 | 2392 | 0.72 | 0.64 |
| 2008 | 1747 | 1466 | 2319 | 0.75 | 0.63 |
| 2009 | 1752 | 1354 | 2301 | 0.76 | 0.59 |
| 2010 | 1799 | 1327 | 2285 | 0.79 | 0.58 |
| 2011 | 1797 | 1288 | 2257 | 0.8 | 0.57 |
| 2012 | 1809 | 1160 | 2242 | 0.81 | 0.52 |
| 2013 | 1830 | 1113 | 2235 | 0.82 | 0.5 |
| 2014 | 1857 | 1048 | 2221 | 0.84 | 0.47 |
| 2015 | 1838 | 994 | 2151 | 0.85 | 0.46 |
| 2016 | 1791 | 916 | 2081 | 0.86 | 0.44 |
| 2017 | 1629 | 835 | 2026 | 0.8 | 0.41 |
| 2018 | 1730 | 772 | 1973 | 0.88 | 0.39 |
| 2019 | 1705 | 708 | 1909 | 0.89 | 0.37 |
| 2020 | 1686 | 650 | 1881 | 0.9 | 0.35 |
| 2021 | 1679 | 603 | 1839 | 0.91 | 0.33 |
| 2022 | 1658 | 534 | 1796 | 0.92 | 0.3 |
| 2023 | 1533 | 455 | 1657 | 0.93 | 0.27 |

since the mid-2000s, exceeding the stock option ratio for the first time in 2006, reaching 0.8 in 2011 and above 0.9 in 2020. During the same period, the stock option ratio continued to decline, reaching only 0.27 in 2023, being replaced by RSUs.

According to another study, the increase in RSUs in Korea is not notable, but the utilization of stock options is decreasing as in other countries. The number of people who received stock options in Korea decreased by about 33% from 16,087 people in 2021–10,835 people in 2023. This shows that Korea also recognizes the limitations of the existing stock option system as a means of compensation, such as short-term performance and bias, and moral problems when setting exercise prices [10].

Stock-based compensation, including RSUs, appears to have different effects depending on the structure and environment of each industry. In technology-intensive industries, due to their high dependence on human capital and high growth potential, the proportion of stock-based compensation to secure key talents is relatively high [50]. Otherwise, in industries with high regulatory intensity, such as financial and public services, there is a strong tendency to operate stock compensation conservatively due to the high risks associated with capital regulation, accounting regulation, and conflict of interest control. This difference suggests that operating the same RSUs plan can act as a structural factor that spreads and settles across the organization in some industries but still stays at an early level in other industries [42,51].

Recent quantitative studies are analyzing how incentive-performance relationships differ by industry characteristics. Jia and Che [52] analyzed that the stability of institutional investor shares for Chinese listed companies improves corporate innovation through managerial stock incentives, and this effect shows heterogeneity that varies depending on corporate characteristics such as managerial control level and company size. Kayani and Gan [51] demonstrated the relationship

between managerial remuneration and financial performance using data from companies in the Asia-Pacific region, showing that the pay-performance relationship differs depending on the market and institutional environment in which the company belongs. In a study that estimated the value of restricted stocks granted to non-executive employees, Abudy and Benninga [53] argued that an average discount of about 30% occurs even for stock compensation of the same nominal value due to non-marketability and non-diversification constraints, and that the degree of discount can vary greatly depending on the characteristics of the company and industry. Oxelheim et al. [54] presented evidence that CEO remuneration in Europe responds sensitively to macroeconomic and industrial-level shocks along with corporate performance, emphasizing that the analysis of the management compensation system should consider not only the internal characteristics of the company but also the environment in which each industry faces.

Park et al. [4] analyzed the impact of the introduction of RSUs by U.S. listed companies on EPS and operating profit by industry. In technology-intensive industries such as IT, which have a high proportion of growth potential and intangible assets, improvements in EPS and operating profit growth are seen immediately after the introduction of RSUs, but in industries with highly regulated and capital-intensive industries such as manufacturing and energy, patterns of relatively delayed or small scale are observed.

Kim and Yang [10] also analyzed U.S. listed companies that introduced RSUs, pointing out that RSUs issuance has a significant positive effect on sales in manufacturing, information technology, healthcare, telecommunications, materials, and energy industries, but statistical significance was not observed in the financial, utility, and real estate industries, which differed by industry. And Lee [19] said that in the manufacturing, financial and public sectors in Korea, law, institutional environment, strong labor-management relations, and conservative recognition of shareholder value act as factors limiting the introduction of RSUs, and that even within the same country, different industries show different patterns.

## 2.5. Strategic management process and RSUs

The 'Strategic Management Process' is a widely used management framework and generally consists of three stages: strategy setting, strategy execution, strategy evaluation/control [55–57].

The organization's mission and vision are clarified in the strategy setting stage, opportunities and threats, and strengths and weaknesses of internal capabilities are analyzed, and specific strategic goals are established. The established strategy is then embodied through organizational structure design, resource allocation, leadership placement, organizational culture alignment, etc., and in the evaluation and control stage, the implementation result is quantitatively and qualitatively monitored, and the strategy is supplemented and coordinated through feedback.

The strategic management process was developed based on classical strategic theories like Ansoff, Chandler, Mintzberg and Porter [58–61]. Ansoff [58] systematized strategic planning and execution into organic and continuous activities. Chandler [59] emphasized the alignment of strategy and organizational structure as the representative proposition of "structure follows strategy" shows. Mintzberg [60] suggested the need for strategic flexibility, saying that strategies can gradually evolve during implementation rather than just planning and Porter [61] underscored the importance of establishing a systematic strategy to respond to the external environment based on industrial structure analysis.

The strategic management process gradually goes beyond a plan-oriented approach and develops into an integrated framework that encompasses learning, strategic evolution, and dynamic adaptation to the external environment, and is applied to management models in various areas such as public policy, non-profit organizations, and human resource management as well as corporate management [62].

From a strategic management perspective, RSUs are a strategic incentive mechanism to secure key talents and link long-term performance with shareholder value [4,5]. Therefore, RSUs must transcend simple personnel policy and be integrally managed within the corporate strategy framework to ensure their systematic operation [6].

In this context, the strategic management process can function as a framework for managing the life cycle of the RSUs system. The strategic management process can secure operational consistency and institutional control by structuring the progressive stages such as design, assignment, performance linkage, evaluation, and supplementation of RSUs. The direction of the institutional design is determined in the strategy establishment stage, the organization's execution capability is secured in the execution stage, and a cyclical structure that verifies and improves effectiveness is required in the evaluation stage. This logic is linked to classical strategic theories, where Ansoff's plan-oriented approach reinforces the systemicity of RSUs structural design [58] and Chandler's strategy-structure alignment theory is the basis for institutionalization within the organization [59]. Mintzberg's concept of strategic evolution suggests that flexibility should be secured in the operation process of RSUs [60]. Porter's analysis of the industrial structure can emphasize the need for institutional design tailored to the external competitive environment and industrial characteristics [61] and Hunger and Wheelen revealed that the strategic management process can increase environmental adaptability and institutional coordination through performance monitoring and feedback as well [55].

Recent discussions in strategic management are evolving in the direction of simultaneously creating long-term value for companies and achieving organizational sustainability [63,64]. This change is not a 'new trend', but is also the result of a rapid emergence of the importance of human capital management as a key element of strategy execution. In order to introduce and operate RSUs as a strategic device that maintains key talents for a long time, promotes organizational commitment, and strengthens the companies sustainable growth foundation, the RSUs must be closely linked to the company's overall strategic goals (especially long-term growth, responsible management, and human capital accumulation) in connection with the strategic management process so that the RSUs system can ultimately act in the direction of enhancing corporate strategic flexibility and organizational resilience.

## 2.6. Delphi research methodology

**2.6.1. Theory and key concepts of Delphi.** 'Delphi' originates from the oracle at the ancient Greek temple of Delphi. Kaplan introduced the basic concept at the RAND Corporation in the 1940s, and it was refined into a methodology by Dalkey and Helmer, Gordon and Helmer when applied to long-term forecasting problems in military and security fields [64–66]. The US Department of Defense's use of Delphi for nuclear attack damage prediction projects was the decisive moment for the spread of Delphi methodology and has since been used in various areas such as policy development, science and technology roadmap, and organizational strategy design beyond the defense sector [67].

Delphi is a structured decision-making methodology designed to approach reasonable conclusions by systematically collecting the experiences and judgments of a group of experts in problems that are complex and difficult to define a single correct answer due to high uncertainty [68]. Delphi has academic and practical significance in that it repeatedly collects and refines the opinions of relevant experts on topics and issues that are difficult to explain with quantitative indicators to reach a convincing agreement [69].

The Delphi technique has four main characteristics: anonymity, repetitive surveys, controlled feedback, and expert consensus [69]. By not disclosing the identity of the participants, it reduces the likelihood that a particular authority or hierarchy within the organization affects the response, and conducts multiple surveys, each expert is asked to review their views by referring to the group's response distribution. The level of consensus finally reached is considered a collective judgment that reduces the likelihood of errors compared with that of a single expert. Because it originated in Greece, where democracy developed, democratic decision-making principles and quantitative rationality lie on the theoretical basis of the Delphi methodology, the premise that majority judgments are more reliable than individuals, and the assumption that collective intelligence can exercise statistical wisdom in uncertain situations [70].

**2.6.2. Advantages of the Delphi.** Delphi methodology is consistently used in many fields because it can be more effective than conventional face-to-face meetings or one-time surveys. Delphi has the effect of reducing costs [71]

because even experts who are geographically difficult to attend can participate in a non-face-to-face manner, which is useful when constructing panels for international research topics.

Delphi alleviates the pressure to agree with authorities or majority opinions by ensuring anonymity. Since respondents participate without revealing their name or affiliation, they can present their judgments without being conscious of the hierarchy or personal relationships within the organization, leading to an honest and independent response [69,72].

Delphi's iterative questionnaire and controlled feedback structure also works as a device to promote opinion gathering. After each round, researchers provide summary statistics such as mean, median, and standard deviation for each item, and experts can refer to this to maintain or adjust their responses to observe the tendency of the level of consensus to increase as the variance of group responses gradually decreases [73,74]. The Delphi technique prevents errors in groupthink by structurally blocking situations in which a specific individual or minority leads the discussion. In each round, experts refer to group statistics, but can adjust their own judgments without having to directly argue or persuade other experts, and this circular structure of 'independent judgment-group information-re-adjustment' is an important advantage of Delphi [72,74]. And Delphi results show not only majority opinions but also minority opinions and items that are difficult to agree on, and the researcher can describe the scope and limits of the agreement at the same time based on this [68].

**2.6.3. Delphi research process.** The Delphi research process can generally be organized into several steps. The first task to do is to clearly define the problems and research questions that the researcher wants to solve through literature research, and then gather experts with abundant knowledge and a lot of experience on the research topic. In Delphi, panel composition is a factor that determines the quality of research, and representation, professionalism, continuity and sincerity of responses are essential, and existing studies generally view a panel size of 10–30 as appropriate [72].

The starting round is usually conducted as an open-ended questionnaire. The purpose of this stage is to discover a wide range of opinions, indicators, and factors as much as possible, and focuses on grasping what experts recognize as it is rather than forcibly fitting into the preset category by the researcher [73]. The collected responses integrate similar items through content analysis and remove duplicates.

In the next round, a closed questionnaire will be conducted with organized items. Statistics like average and median values are calculated by evaluating the importance, suitability, and feasibility of each item using the Likert scale and presented as feedback data to experts [72,74]. Experts choose whether to maintain or modify their responses by looking at their responses and group statistics. This process stabilizes the distribution of responses, and the researcher determines the end of the investigation according to the pre-set consensus criteria [75,76].

In the later round, the results are interpreted by synthesizing the level of agreement, residual disagreement, and response changes for each item. The Delphi study also considers the fact that lack of consensus on a specific item is also important information and presents this part as a 'controversial area' that requires further research or policy discussion in the future [67,77]. To ensure rigorous and objective research, Delphi results are sometimes statistically validated to establish reliability and validity.

In this study, a three-round Delphi survey was conducted with 31 experts (11 from the legal and accounting group and 20 from the strategy, HR, and IR group) to derive decision-making items for each stage of the proposed RSUs framework (strategy setting, execution, evaluation, and control). The first round collected open-ended responses, which were coded and consolidated through content analysis by merging overlapping statements and removing duplicates to form the candidate item pool. The refined items were then evaluated in the second and third rounds using Likert-scale ratings, and the final level of content validity and agreement was statistically verified using CVR and Kendall's W. The Delphi research method procedure described above can be summarized as shown in 'Fig 1' below.

**2.6.4. Delphi application cases and trends.** Delphi methodology is a steady seller utilized across various academic and industrial fields. In the highly prominent field of artificial intelligence (AI), the Delphi technique is employed as one tool for forecasting technological development paths, regulatory directions, and socioeconomic ripple effects. Alon et al. [78]

| **Preliminary Study** | - Identify candidate priority items through literature review and prior studies<br>- Select expert panel<br>- Draft first-round open-ended Delphi questionnaire |
| --- | --- |

| **First-round Delphi Survey**<br>(Open-ended Questionnaire) | - Distributed the first-round open-ended Delphi questionnaire to the expert panel. |
| --- | --- |

**Feedback to Experts**

| **Additional-round Delphi Survey**<br>(Closed-ended Questionnaire, Assessment of Item Importance) | - Distributed the second-round Delphi questionnaire<br> (If necessary, a third-round survey is also conducted)<br>- Analyzed the importance of each item in the survey<br>- Verified validity and reliability |
| --- | --- |

**Fig 1. Delphi research method process.**

reviewed AI-related Delphi studies conducted since the 2010s, summarizing research that refined AI ethical norms, the employment impact of automation, and data governance principles through expert consensus.

Arribas et al. [72] derived the necessary digital competencies and educational content for designing an education and training system to counter AI-based disinformation and information manipulation in Europe through expert panel consensus. In a subsequent paper by the same research team, based on the same Delphi panel and survey, they detailed a methodological design for early identification of AI-assisted disinformation detection capabilities and related curriculum requirements, providing a concrete model for Delphi application linked to a digital competency framework [76].

In the healthcare sector, Delphi has been employed in developing clinical practice guidelines, patient safety indicators, and service quality assessment criteria. Khodyakov et al. [77] systematically reviewed Delphi studies in this domain, proposing consensus recommendations for standardizing research design and reporting.

Palomino Pichihua [79] performed a Delphi survey to derive major issues and policy priorities that may arise in the process of introducing smart cities and proposed a step-by-step implementation strategy. Kavoura and Andersson [80] applied the Delphi methodology to design the strategy of social enterprise, and repeatedly adjusted expert opinions to derive the strategic components of the social enterprise model. Feng et al. [81] identified strategic human resource management capabilities using the Delphi methodology for clinical departments of public hospitals in China and established key competency indicators required for HRM design of hospital organizations. Rakowska and Cichorzewska [82] repeatedly collected the opinions of various industry experts to derive the technical, social, and cognitive capabilities that will be required in the future labor market.

## 2.7. AHP research methodology

### 2.7.1. Theory and key concepts of AHP.
AHP is a tool designed to help rational choices by organizing complex and multidimensional problems with the 'Multi-Criteria Decision Making' (MCDM) methodology. AHP was developed in the early 1970s by Professor Thomas L. Saaty of the University of Pennsylvania in the United States to improve the inefficient decision-making process experienced during the State Department's arms control and arms reduction negotiations

[83,84]. AHP methodology should layer the elements of decision-making problems to meeting the MECE (Mutually Exclusive, Collectively Exhaustive) principle. Elements belonging to the same layer should not overlap each other (Mutually Exclusive) and at the same time, Collectively Exhaustive [85].

The key point of the AHP procedure is the process of quantifying the relative importance by pairing elements of the same layer one-on-one. The evaluator compares the importance of the two elements using a 1–9 points scale and organizes them into a pairwise comparison matrix using eigenvalues and eigenvectors to calculate the relative weight of each element [83,86], which allows the evaluator's intuition to be statistically verified by quantifying the results [84].

**2.7.2. Advantages of AHP methodology.** By decomposing decision-making problems into hierarchical structures such as 'goals-evaluation criteria-alternatives', the AHP methodology enables a view of the entire structure at a glance and makes it easy to derive key issues in multi-faceted problems [87].

Unlike traditional numerical-based analysis, AHP is characterized by being able to convert qualitative judgments including intuition, experience, and emotion of experts into quantitative information [88]. Zahhedi [89] defined AHP as "a methodology that enables comprehensive problem solving by reflecting the evaluator's intuitive judgment and rational thinking while considering multiple alternatives and multiple evaluation criteria together in complex situations." Because of these characteristics, the AHP methodology is particularly useful in environments where data are incomplete or have high uncertainty [84]. Experts can make comprehensive decisions that take into account quantitative and qualitative information together because qualitative information is converted into numerical data in the process of recording the results of pairwise comparisons of elements based on their own experience and intuition, which can be used in situations where it is difficult to predict the future or secure enough data [87].

Another important feature of AHP is that it simplifies the complex judgment process. In AHP, it is designed to compare only two alternatives at all times rather than multiple alternatives at once, reducing the cognitive burden on the evaluator. This reduces judgment errors and enables relatively consistent evaluation even in situations where various criteria and alternatives coexist [84]. AHP analysis can be performed with consistency verification to ensure the reliability of the evaluation results. AHP calculates the Consistency Index (CI) and the Consistency Ratio (CR) to confirm how logically consistent the evaluator's judgment is. Here, CR is an indicator of the logical consistency of the pairwise comparison matrix and is a value obtained by dividing CI by random index (CR = CI/RI), and a smaller value means greater consistency [90]. The CR value of 0.1 or less is a sufficiently consistent judgment and 0.2 or less is considered an acceptable level [90].

If the CR value is greater than or equal to 0.1 or multiple issues collide, sensitivity analysis can be used to further check the robustness of the results as needed. Sensitivity analysis is not an essential procedure, but rather a complementary verification tool that can be selectively performed in consideration of research objectives and data constraints. If the priority of the alternative does not change significantly when the weight or input value of the evaluation criterion is changed, the derived weights and rankings can be interpreted as relatively stable, and this procedure can be used in decision-making situations that are very sensitive to change in results [84,90].

**2.7.3. AHP research process.** The first step in AHP research is to specify the problem to be solved and then apply the MECE principle so that the elements of each layer can sufficiently explain the higher goals without overlapping each other for each problem-solving process [87]. The next step is pairwise comparison. The 1–9 points scale (Table 2 below) is used to give relative importance by comparing elements located in the same layer with each other. One point means that two elements have the same importance, and nine points mean that one element is extremely more important than the other [90].

The method of comparing only two elements (Table 3 below) simplifies complex judgments to reduce the burden on evaluators and increase consistency. The collected comparison results are organized into pairwise comparison matrices and serve as basic data for mathematical analysis.

Next is the Weight Estimation step, and the calculated weight represents the relative importance of each criterion and alternative. The relative weight of each element is derived by calculating the eigenvectors with a pairwise comparison

**Table 2. Saaty's 9-point scale [90].**

| Importance | Definition | Content |
|---|---|---|
| 1 | Equal Importance | Based on certain criteria, two activities are judged to have similar contribution levels |
| 3 | Moderate Importance | Based on experience and judgment, one activity is slightly preferred over another activity |
| 5 | Strong Importance | Based on experience and judgment, one activity is strongly preferred over another activity |
| 7 | Very Strong Importance | Based on experience and judgment, one activity is very strongly preferred over another activity |
| 9 | Extreme Importance | Based on experience and judgment, one activity is strongly extremely preferred over another activity |

**Table 3. Example AHP survey items.**

| Criteria | Absolute Important | Very Important | Important | Slightly Important | Similar | Slightly Important | Important | Very Important | Absolute Important | Criteria |
|---|---|---|---|---|---|---|---|---|---|---|
| Importance | 9 | 7 | 5 | 3 | 1 | 3 | 5 | 7 | 9 | Importance |
| Item1 | | | | | | | | ✓ | | Item2 |

matrix, and CI and CR are calculated to evaluate the consistency of the results, and if the CR is less than 0.1 it is considered that sufficient consistency has been secured [90].

The final step is weight aggregation, which combines the weights obtained for each layer in a multiplication method between the upper and lower layers, and finally calculates the weights of each alternative to derive the priorities for each alternative and compares and selects items based on a mathematical basis [85].

In this study, AHP was applied to prioritize the Delphi-derived decision items across four GICS-based industry groups (consumer goods; resources and energy; industry and infrastructure; technology and communications). A total of 48 leaders (12 per industry group), including C-level, executives, and team leaders, completed pairwise comparisons using Saaty's 1–9 scale, and reciprocal comparison matrices were constructed for weight estimation. Consistency was assessed using CI and CR, and only responses meeting the conventional threshold (CR < 0.1) were used to aggregate weights across hierarchy levels and derive final priorities. The AHP process described above is summarized in a diagram as 'Fig 2' below.

**2.7.4. AHP application cases and trends.** Prior to 2000, AHP methodology was mainly used in facility investment or project selection problems in traditional industries such as manufacturing, construction, and transportation [85], but it has recently been used as an important decision-making tool in emerging areas such as healthcare, IT, energy, smart city, and environmental management.

Aragonés-Beltrán et al. [91] combined AHP and an Analytical Network Process that performs a pairwise comparison to reflect the interdependence and feedback between decision-making factors to determine the priority of investment in photovoltaic projects in Spain. This study shows that complex investment alternatives can be evaluated more realistically and shows that an AHP-based approach can support decision-making even in high-risk and large-scale investment situations. Mbuli [92] analyzed the status of wind power generation with AHP and confirmed that AHP works as a major decision-making tool even in setting standards for detailed areas such as R&D investment, site selection, and equipment purchase.

Improta et al. [93] conducted a study to improve the process of the emergency department of Cardarelli Hospital in Italy by combining AHP with Lean Thinking [94], an innovative philosophy that aims to eliminate waste by leaving only essential items in the Health Technology Assessment, making it faster and better with fewer resources.

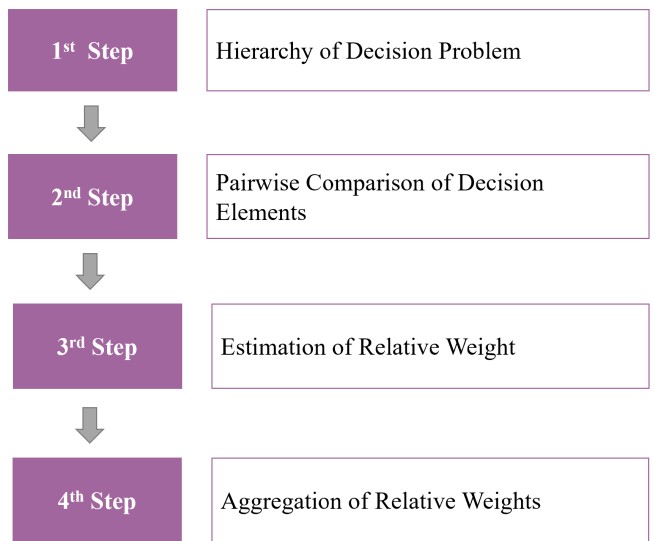

**Fig 2. AHP research process.**

In the human resource management (HRM), Salehzadeh and Ziaeian [95] conducted a literature review, summarizing that MCDM techniques like Fuzzy AHP are widely used for core HRM challenges such as talent selection, promotion evaluation, and team composition optimization. They emphasized that AHP is a practical decision-support tool in modern HR strategy development. In transportation and logistics sector, Macharis and Bernardini [96] assessed that techniques like AHP are well-suited for the Multi-Actor Approach, which integrates the opinions of diverse stakeholders in transportation projects.

AHP methodology utilization is also active in engineering and agriculture. Almeida et al. [97] integrated the 'multi-criteria and multi-objective optimization model' [98] – which simultaneously considers multiple evaluation criteria and objectives to find a set of 'compromise solutions' rather than a single optimal solution – with AHP. This demonstrated its applicability in industrial risk management, equipment reliability assessment, and maintenance optimization. Oliveira and Duarte [99] proposed Automatic AHP and Semi-automatic AHP to enhance the efficiency of agricultural decision-making. They addressed the limitations of speed and consistency inherent in traditional manual AHP and displayed its potential for large-scale application in agricultural investment and crop selection.

## 2.8. Delphi-AHP research methodology and application

Delphi and AHP have their own strengths, but there are areas that need to be supplemented. The Delphi methodology can systematically collect expert collective knowledge, but there is room for the subjectivity of researchers to be involved in the process of constructing the panel, and the quality of responses may deteriorate as experts feel tired due to repeated surveys. Moreover, when the disagreement within the panel is large, it takes a lot of time to reach an agreement, and there is a possibility that groupthink may occur as feedback is repeated too much above the appropriate level [100]. The AHP methodology can quantify the relative importance of each factor through pairwise comparison, but due to human cognitive limitations, it is difficult to maintain consistency in judgment as there are more pairwise comparisons [101].

Delphi-AHP methodology, which combines Delphi and AHP, complements these limitations. This methodology can secure consistent and reliable input data by using the resulting items and criteria derived from collecting expert opinions through Delphi as a pairwise comparison element of AHP [102]. In other words, Delphi is responsible for refining and

structuring expert judgment, and AHP provides an integrated framework leading to an analysis stage in which it converts into quantitative weights and priorities.

Delphi-AHP methodology can be found in studies in various fields. Di Zio and Gordon [103] sought a method of combining Nudge Theory (Sugden, 2009 [104]) to Delphi-AHP. By applying the nudge theory to AHP, which induces more desirable choices with an environmental design that does not limit freedom of choice, it showed the effect of increasing consistency of expert judgment and reducing cognitive bias.

Anastasiadou et al. [102] took the example of Thessaloniki, Greece, and conducted a Delphi survey on institutional, financial, social, and technical barriers in the process of promoting sustainable urban transportation policies, and then used AHP to quantify the relative importance of each barrier. As a result, the lack of coordination between agencies, weak governance, and lack of budget were identified as key obstacles, and the researchers presented this as a policy decision-making tool that can be used in the early stages of urban transportation policy. Han [105] used Delphi-AHP to evaluate the priorities of industrial and national policies surrounding the introduction of AI. Using technology readiness, economic ripple effects, social impact, and policy feasibility as evaluation criteria, expert consensus was reached through Delphi, and AHP weights were estimated to present manufacturing and health care as priority support industries, and AI infrastructure and education investment as key policy areas.

In the HR field, Darvazeh et al. [106] derived the eco-friendly HRM elements of the construction industry as Delphi and analyzed the priorities of recruitment, compensation, education, delegation of authority, and environmental management with AHP to show that eco-friendly recruitment and eco-friendly compensation are key to the sustainability of construction companies. Matemane et al. [107] utilized Delphi-AHP to examine which ESG indicators should be reflected first in the executive compensation design of companies in emerging countries, and confirmed that the environment was the highest priority, and that waste and emission management indicators were key factors as detailed items. Rizky et al. [108] contributed to enhancing the fairness and transparency of the public sector reward system by refining the evaluation criteria necessary for the government agency's outstanding employee reward system with Delphi and calculating the weights with AHP, presenting discipline and innovation as the top evaluation factors.

In the field of quality management and safety and health, Nguyen et al. [109] conducted a Delphi survey of manufacturing and management experts in the process of building the TQM 4.0 model to derive key indicators, and confirmed that CEO commitment, digital integration, and quality culture are the highest priority factors with AHP. Kosolsaksakul and Ketsakorn [110] selected and refined the requirements for introducing the ISO 45001 safety and health management system of small organizations with Delphi and reported that leadership and worker participation were the top application areas with the AHP methodology. Khan et al. [111] systematically classified and prioritized structural obstacles in procurement and supply chain management of public sector projects in developing countries as Delphi-AHP, and Borges et al. [112] utilized Delphi-AHP to establish an expert consensus-based evaluation system to evaluate biomass sustainability in the sugar industry.

## 3. Research methodology

### 3.1. Delphi methodology and data

RSUs are a mechanism in which multiple factors interact, including compliance with legal norms, accounting treatment, talent acquisition strategy, and stakeholder communication [4–6]. To investigate the RSUs system from various perspectives, a panel was formed of experts from law, accounting, strategy, HR, and IR, which are key positions related to RSUs introduction and operation.

In legal and accounting, both interpret and apply normative systems such as commercial law, capital market law, taxation, and IFRS, and perform the function of 'oversight and compliance' to check the adequacy of financial reporting, integrity of disclosures, and compliance with regulations. They perform 'oversight and compliance' functions, verifying the appropriateness of financial reporting, the completeness of disclosures, and regulatory adherence. These professions

are conceptualized in existing corporate governance research as part of the same gatekeeper group alongside auditors and legal advisors. They act as third-party enforcers ensuring regulatory compliance and information reliability, thereby mitigating information asymmetry between companies and investors [113–115]. From this perspective, these roles share expertise in comprehensively reviewing the legal validity of system implementation and the appropriateness of accounting treatment, so both groups were integrated in the Delphi study design.

Strategy, HR, and IR roles focus on how RSUs, as a long-term incentive system, can be utilized to execute corporate strategy, retain key talent, and manage shareholder relations. Strategic management and strategic human resource management research views the core task as aligning personnel and compensation policies with the company's competitive strategy. It has also analyzed the mechanisms for building human capital and organizational capabilities through HR systems and linking them to performance [116,117]. The IR function is also discussed as a strategic communication function that explains the company's strategy, compensation, and governance to the capital markets and reflects investor feedback into strategy and institutional design. It is part of strategic management activities that manage corporate value and the investor base [118]. Considering this interconnection and decision-making logic, grouping strategy, HR, and IR together is not only theoretically consistent for capturing the strategic utilization context of RSUs but also aligns with the need to encompass strategic linkages.

The number of experts and their qualification requirements were determined by referencing prior Delphi research. Dalkey [65] and Rowe and Wright [68] suggest that expert groups of 5–20 members can produce reliable results, while Arribas et al. [72] report that even 10–15 members can yield empirically meaningful outcomes. The purpose of Delphi is not to generalize the sample or derive statistical results, but to theoretical structuring through expert consensus [119]. For this reason, existing studies recommend that the size of experts between 10 and 30 with abundant practical experience and judgment skills that can be theoretically structured on a particular problem or topic is an appropriate level for the Delphi expert group [68,75,120]. Based on this, this study prioritized expertise and response consistency over sample size expansion when forming the panel, ultimately selecting 31 members (11 in Legal/Accounting, 20 in Strategy/HR/IR). No experts dropped out during the three rounds of the Delphi process; all 31 panelists participated in each round.

The Legal/Accounting group comprised 11 experts with experience handling legal affairs, compliance, finance/accounting, and disclosure/audit responses at listed companies and large corporate groups. These individuals are executives and C-level leaders holding certified professional qualifications such as attorneys and Certified Public Accountants (CPA), with experience across multiple industries including Shipbuilding and Energy, Energy and Machinery, Chemicals and Energy, Construction and Machinery, Machinery and Chemicals, Shipbuilding and Defense, Hotels and Distribution, Construction and Materials, Aerospace, Electronics and IT. The list of the Legal and Accounting group is shown in 'Table 4' below.

The Strategy·HR·IR Group consists of 20 executives and operational leaders responsible for corporate strategy, human resources and compensation, and investor relations, as shown in 'Table 5' below. These leaders come from a wide range of industries including chemicals/energy, defense/energy, energy/construction, electronics/IT, aerospace/aviation/ shipbuilding, construction/chemicals, chemicals/automotive, hotels/distribution, shipbuilding/ energy/defense, materials/ energy, aerospace/aviation/defense, automotive/aerospace, and electronics/IT/communications. A significant number are leader-level and hold C-level, executive, or professional qualifications such as labor attorneys. They interpret the application of RSUs in connection with strategic objectives such as long-term talent retention, enhanced organizational engagement, and increased shareholder value.

The financial industry is discussed as an industry that is generally excluded from the sample in performance comparison or financial analysis because its asset and debt structure, regulatory environment, and profit structure are very different from those of non-financial industries [121]. A study that analyzed the effect of credit risk-related indicators such as interest income, bad debts, and provision for bad debts on EPS and after-tax profits for banks also shows that the performance determinants of the financial industry are structurally different from those of non-financial industries [122]. Although this study is not a financial analysis, financial experts were excluded from the Delphi analysis to ensure consistency in

**Table 4. Legal/accounting professionals.**

| No. | Industry | Experience | Job Description (Qualifications) |
|---|---|---|---|
| 1 | Shipbuilding/Energy | 20 + years | Executive (Lawyer) |
| 2 | Energy/Machinery | | Team Leader (Lawyer, CPA) |
| 3 | Chemicals/Energy | | CFO (CPA) |
| 4 | Construction/Machinery | | Executive |
| 5 | Machinery/Chemicals | | Executive (Lawyer) |
| 6 | Shipbuilding/Defense | 15-20 years | General Manager (Lawyer) |
| 7 | Hotel/Distribution | | |
| 8 | Chemicals/Energy | | |
| 9 | Construction/Materials | | |
| 10 | Space/Aviation | | General Manager (CPA) |
| 11 | Electronics & IT | | |

**Table 5. Strategy/HR/IR professionals.**

| No. | Industry | Experience | Job Description (Qualifications) |
|---|---|---|---|
| 1 | Chemicals/Energy | 20 + years | CEO (Adjunct Professor) |
| 2 | Defense/Energy | | Executive |
| 3 | Energy/Construction | | |
| 4 | Electronics/IT | | |
| 5 | Space/Aviation/Shipbuilding | | |
| 6 | Construction/Chemical | | |
| 7 | Chemical/Automotive | | Team Leader (Patent Attorney) |
| 8 | Chemical/Energy | 15-20 years | Team Leader (Labor Attorney) |
| 9 | Hotel/Distribution | | |
| 10 | | | |
| 11 | Shipbuilding/Energy/Defense | | General Manager |
| 12 | Materials/Energy | | |
| 13 | Space/Aviation/Defense | | |
| 14 | Automotive/Space/Aviation | | |
| 15 | Chemicals/Energy | | |
| 16 | Electronics/IT/Communication | | |
| 17 | Automotive/Energy | | |
| 18 | Electronics/IT | | |
| 19 | Materials/Energy | | |
| 20 | Electronics/Energy | | |

extracting items for introducing and operating RSUs because the responses of the expert group reflect the characteristics of the industry.

The Delphi survey method was executed through in-person visit, email, and phone call in parallel. All participants agreed to participate after hearing explanations on the purpose of the study, scope of use, and guarantee of anonymity of responses. Questionnaire responses were collected and analyzed separately from individual identification information, and responses of specific individuals were not exposed. The reason is to reflect the principles of the Delphi methodology of minimizing authority influence and groupthink and inducing more independent judgments through anonymity and controlled feedback structures [68, 69, 123].

 

The Delphi survey was conducted three times. In the first round, an open-ended survey was conducted on the items necessary for the introduction and operation of RSUs, and the researcher integrated and refined similar items by analyzing the contents of the responses. In the next round, the results of the first round were taken to evaluate whether the items were 'essential' in the framework for RSUs introduction, operation, and management, and the validity was verified by Lawshe's Content Validity Ratio (CVR) [124]. The formula of CVR is as follows.

$$CVR = \frac{Ne - (N/2)}{(N/2)}$$

Ne: Number of experts who rated the item as 'essential'

N: Total number of experts

Lawshe [124] assumed a 50% probability that experts would randomly respond that a specific item was 'essential'. He then presented the minimum CVR threshold required to achieve statistical significance at a 5% significance level ($p < 0.05$), as shown in 'Table 6' below. CVR values become stricter and higher as the number of Delphi participating experts decreases, while statistical significance can be achieved with relatively lower CVR values as the number of experts increases. This is based on the logic that the expected probability of random agreement varies with sample size, necessitating stricter criteria for agreement levels in smaller groups.

However, it has been pointed out that Lawshe's CVR criterion alone is insufficient to adequately control for the possibility of chance agreement. To address this limitation, this study adopted the CVR threshold proposed by Ayre and Scally [125]. This threshold is based on Lawshe's theory but recalculates the CVR value using a binomial distribution as described by Wilson et al. [126]. This approach aims to ensure greater reliability in the research results. The corresponding CVR threshold values are also summarized in 'Table 6'.

In the third round, experts were asked to rate importance of the items from the second Delphi results using a 7-point Likert scale ranging from 1 (not at all important) to 7 (very important). Items that passed the CVR threshold but exhibited a standard deviation of 1.5 or higher in the importance assessment—indicating significant opinion dispersion—were excluded from the results, as they were judged to have many conflicting responses [119,127]. This reflects existing discussions that it is preferable to separate items with highly divergent opinions into 'contested issues' or exclude them from the final consensus list, as they are unlikely to represent consensus in Delphi studies [128]. Consequently, only items meeting both the CVR threshold and a standard deviation of 1.5 or less remained on the final list after Round 3. This also satisfies the recommendation by Polit and Beck [129] that content validity assessment should consider not only statistical indicators but also supplementary information such as expert judgment and the degree of dispersion.

To ensure higher reliability and objectivity of Delphi survey results, statistical indicators that can quantitatively verify the consistency of evaluations among expert groups are highly beneficial. To this end, this study applied Kendall's Coefficient of Concordance, W (Kendall's W), a representative method used to assess expert consensus levels in Delphi studies, to validate the third-round Delphi results. Kendall's W is a statistical measure that quantifies, on a scale of 0–1, how closely

**Table 6. Minimum CVR standard value.**

| Number of experts | 5 | 10 | 15 | 20 | 25 | 30 |
|---|---|---|---|---|---|---|
| Minimum CVR standard value (Ayre and Scally) | 1 | 0.8 | 0.6 | 0.5 | 0.44 | 0.33 |
| Minimum CVR standard value (Lawshe) | 1 | 0.6 | 0.49 | 0.42 | 0.37 | 0.33 |

the rankings assigned by multiple experts to the same set of items align. Linstone and Turoff [119] noted that Kendall's W is one of the most widely used empirical tools for verifying the level of agreement among panels in Delphi studies. Hsu and Sandford [120] recommended in their Delphi design guide that W values be used to report panel agreement as an objective numerical value rather than a 'subjective impression.' Diamond et al. [128] similarly emphasized in their Delphi study reporting guidelines that researchers must present quantitative indicators like Kendall's W to supplement claims of consensus that are otherwise only described narratively. Kendall's W is generally defined as follows.

$$W = \frac{12 \times S}{m^2(n^3 - n)}$$

W: Kendall's W value indicating the agreement among experts
S: Sum of squared deviations of rank sum
m: Number of experts participating in the evaluation,
n: Number of items being evaluated
The Kendall's W value itself is a continuous measure, but in Delphi literature, it is often presented as a level of agreement within specific intervals for ease of interpretation. For example, the criteria summarized in 'Table 7' below, based on Lim and Park [130], Romero Jeldres et al. [131] are widely used for analysis.

These intervals are closer to practical guidelines than absolute norms, taking into account the research context, panel composition, and number of rounds. In the Delphi study, if it is 0.3 or higher, it is considered to be the level of actual agreement, and if it is 0.7 or higher, it is interpreted that the opinions of experts are very consistently collected. As such, Kendall's W presents the level of agreement of the Delphi panel as a numerical value so that the level of agreement that depended on the subjective judgment of the researcher can be explained as an objectively verifiable statistical basis. This study also attempted to secure the fact that it is a statistically verified consensus structure by calculating Kendall's W value for each process. Romero et al. [131], while comprehensively reviewing the CVR method and subsequent threshold adjustment discussions, emphasize that in the social sciences dealing with small expert panels, practical criteria setting and operation considering research objectives and panel size are necessary.

To summarize the Delphi methodology used in this study: first, only items meeting the CVR criteria were selected; next, items with a standard deviation of 1.5 or higher were additionally removed; finally, level of consensus was verified using Kendall's W for each process. This combined approach of CVR–dispersion–consensus indicators not only ensure the rigor of Delphi results but also aligns with recent recommendations in Delphi methodology.

## 3.2. AHP methodology and data

The AHP analysis of this study utilizes items of the Delphi study results. Items related to legal and accounting are subject to a common normative system, and an integrated analysis was performed without classifying by industry in consideration

**Table 7. Value Range of Kendall's W criteria.**

| Value Range | Level of Agreement |
|---|---|
| 0~0.1 | Very low agreement |
| 0.1~0.3 | Low agreement |
| 0.3~0.5 | Moderate agreement |
| 0.5~0.7 | Strong agreement |
| Over 0.7 | Very strong agreement |

of the features of comprehensively reviewing the adequacy of a company's financial reporting, completeness of disclosure, and regulatory compliance.

Strategy, HR, and IR differ significantly across industries. Factors like capital intensity, cyclical sensitivity, workforce structure, and R&D proportion lead to differing demands and expected effects for long-term incentive systems such as RSUs. Consequently, the purpose and methods for introducing and operating RSUs can vary substantially by industry [10,41,50]. Therefore, this study conducted an AHP analysis using industry groups classified based on the 'Global Industry Classification Standard' (GICS) jointly developed by MSCI and S&P Dow Jones Indices [4]. The industry classifications are as shown in 'Table 8' below.

This study aims to analyze Groups 1–4 in the above 'Table 8'. As of May 1, 2025, 778 companies (approximately 94%) out of 832 KOSPI-listed firms on the Korea Exchange (KRX) and 1,433 companies (approximately 87%) out of 1,657 KOSDAQ-listed firms belong to these four industry groups. Given this proportion, these four industry groups can ensure representativeness and comprehensiveness of Korean industries and markets, enabling this study to achieve its goal of presenting a methodology and framework for introducing RSUs in Korea [4]. The expert panel for the AHP analysis was defined as senior management overseeing HR/compensation, organizational/business strategy, and investor relations at listed companies and large unlisted firms (within Korean top 500 by revenue, as of 2024) within these four sectors. Specifically, panel members were selected from C-level executives and other leadership-level personnel responsible for HR, strategy, and IR functions, who possessed a minimum of 15 years of experience in their respective fields. The selected expert group is shown in 'Table 9–12' below. As seen in the table, experts are evenly distributed across each function within each industry group, totaling 48 members (12 per group). No participants dropped out during the AHP survey; all 48 panelists completed the pairwise comparisons.

The expert panel size aligns with the recommended standards of existing AHP studies. By setting the size to 12 members per industry group, the problem of specific industries being under- or over-represented was minimized. Saaty [89] suggested that meaningful analysis is possible even with a panel of at least 5 members in group AHP, and presented a range of 7–15 members as a practically desirable size. Studies summarizing other AHP methodologies also suggest that forming a small group of experts with expertise and response consistency is more important than securing an excessively large number of respondents. They indicate that approximately 10–12 members is appropriate for simultaneously ensuring statistical stability and diversity of opinion [83,85]. Furthermore, it has been repeatedly emphasized that in AHP, the logical consistency and expertise of each respondent's judgments are more important than the sample size [90,95]. It has also been noted that an excessively large number of respondents can lead to an exponential increase in the number of pairwise comparison matrices required, causing fatigue and potentially increasing inconsistency [92].

The AHP survey was conducted in person. The reason is the explanation of the AHP methodology, and the understanding of the survey method may differ for each expert, and there are some experts who are experiencing AHP for the first time, so the researcher visited and carried it out in person to increase the understanding regarding AHP method of the experts. The comparison matrix questionnaire of items by process in each industry group. The pairwise comparison

**Table 8. GICS-based industry classification [4].**

| Category | Category Detail | Entry |
|---|---|---|
| Group 1 | Consumer-Related Industries | Consumer Staples, Consumer Discretionary |
| Group 2 | Resource and Energy-Related Industries | Energy, Materials, Utilities |
| Group 3 | Industrial and Infrastructure-Related Industries | Industrials, Real Estate |
| Group 4 | Technology and Communication-Related Industries | Information Technology, Communication Services |
| Group 5 | Health Related Industries | Health Care |

**Table 9. AHP targets by industry group 1.**

| NO | Section | Experience | Job Description (Qualifications) |
|---|---|---|---|
| 1 | HR | 20 + years | Team Leader (Labor Attorney) |
| 2 | | | Team Leader |
| 3 | | 15-20 years | General Manager (Labor Attorney) |
| 4 | | | General Manager |
| 5 | Strategy/IR | 20 + years | Team Leader (CPA) |
| 6 | | 15-20 years | General Manager |
| 7 | | | |
| 8 | | | |
| 9 | Executive | 20 + years | CEO |
| 10 | | | CMO |
| 11 | | | CSO |
| 12 | | | Division Head |

**Table 10. AHP targets by industry group 2.**

| NO | Section | Experience | Job Description (Qualifications) |
|---|---|---|---|
| 1 | HR | 20 + years | Team Leader (Labor Attorney) |
| 2 | | 15-20 years | General Manager |
| 3 | | | |
| 4 | | | |
| 5 | Strategy/IR | 20 + years | Team Leader |
| 6 | | 15-20 years | General Manager |
| 7 | | | |
| 8 | | | |
| 9 | Executive | 20 + years | CEO |
| 10 | | | COO |
| 11 | | | Division Head |
| 12 | | | |

matrix questionnaire for each process and industry group as provided to experts, and responses were collected, and if the response was omitted or the value of the matrix clearly contradicted, a second response was requested by phone. Previous research recommended that the repetition level should be within 2–3 times, as repeated requests for surveys can lead to expert fatigue and lower response rates. After collecting the questionnaire, the values answered by experts were integrated into the 'Aggregation of Individual Judgment' method, which constitutes a group judgment matrix with a geometric mean, and the responses of individual experts were combined [90,132].

The Consistency Ratio (CR) was based on 0.10 or less, but AHP uses a CR value of 0.10 or less as an acceptable consistency criterion [83,90]. Although some applied studies relax the CR criterion to the level of 0.12 to 0.20 in consideration of the number of judgment items, the complexity of the problem, and expert fatigue, this study applied a CR of 0.10 or less for conservative analysis [123,133,134].

## 4. Delphi results

The Delphi survey for this study was conducted in three rounds. As examined in 'Chapter 2: Literature Review,' the Delphi technique is generally recommended to be repeated two to three times as the optimal level to prevent fatigue effects from excessive repetition while securing a sufficient level of consensus [69,75].

**Table 11. AHP targets by industry group 3.**

| No | Section | Experience | Job Description (Qualifications) |
|----|---------|------------|-------------------------------|
| 1 | HR | 20+years | Team Leader |
| 2 | | 15-20 years | |
| 3 | | | |
| 4 | | | General Manager (Labor Attorney) |
| 5 | Strategy/IR | 20+years | Team Leader |
| 6 | | | |
| 7 | | 15-20 years | General Manager |
| 8 | | | |
| 9 | Executive | 20+years | CMO |
| 10 | | | CHRO |
| 11 | | | CSO |
| 12 | | | Division Head |

**Table 12. AHP targets by industry group 4.**

| NO | Section | Experience | Job Description (Qualifications) |
|----|---------|------------|-------------------------------|
| 1 | HR | 20+years | Team Leader |
| 2 | | 15-20 years | General Manager |
| 3 | | | |
| 4 | | | |
| 5 | Strategy/IR | 20+years | Team Leader |
| 6 | | | |
| 7 | | 15-20 years | General Manager |
| 8 | | | |
| 9 | Executive | 20+years | CSO |
| 10 | | | |
| 11 | | | Division Head |
| 12 | | | |

## 4.1. The first Delphi survey results

The first survey was conducted as an open-ended questionnaire. Experts in the legal/accounting and strategy/HR/IR fields were asked to describe items to consider when introducing and operating RSUs according to the strategic management process framework presented in Chapter 2, organized into three stages: 'Strategy Setting', 'Strategy Execution', and 'Evaluation and Control'. The collected responses were consolidated and refined by integrating similar or redundant expressions. For the legal/accounting group, this resulted in a total of 12 items, as shown in 'Table 13' below.

Strategy, HR, and IR groups also consolidated and refined similar or overlapping expressions, resulting in a total of 31 items as shown in 'Table 14' below.

## 4.2. Results and validation of the second Delphi survey

The second Delphi survey was conducted as a closed-ended questionnaire to validate the items derived from the first survey. Experts were asked to evaluate whether each item was 'essential' or not for the introduction and operation of RSUs. The CVR proposed by Lawshe [124] was applied to review the validity of each item. He assumed a 50% probability that an expert would respond that an item was 'essential' and presented a statistically significant minimum CVR threshold

**Table 13. Legal/Accounting items.**

| Process | Initial survey results (18 items) | Duplicate items deleted and consolidated (12 items) |
|---|---|---|
| Strategy Setting | Compliance with the Labor Standards Act, Compliance with capital market laws, Compatibility with international regulations, Compliance with Financial Supervisory Service disclosure regulations, Governance regulations, Applicability to unlisted companies, Necessity of shareholder meeting resolutions, Tax assessment criteria, Tax relief measures, Application of accounting standards, Analysis of country-specific regulations | Compliance with relevant laws and accounting regulations, Applicability to unlisted companies, Necessity of board resolution, Necessity of shareholders' meeting resolution, Tax assessment criteria, Tax relief measures, Analysis of regulations by country |
| Strategy Execution | Information disclosure policy, Handling of RSUs for retired employees, Regulations for handling RSUs upon termination, Preparation for accounting audits and inspections, Income tax filing methods | Information disclosure policy, Post-termination settlement of RSUs, Preparation for accounting audits and inspections, RSUs payment record filing |
| Evaluation & Control | Internal control system establishment | Internal management system operation |

**Table 14. Strategy/HR/IR items.**

| Process | Initial survey results (48 items) | Duplicate items deleted and consolidated (31 items) |
|---|---|---|
| Strategy Setting | Shareholder value maximization model, Design of models tailored to corporate growth stages, Ratio of performance-based RSUs and fixed RSUs, Alignment of performance evaluation with corporate culture, Benchmark setting and comparative analysis, Setting of individual and organizational performance ratios, Performance-based compensation models, Transparency of compensation criteria, Establishment of corporate performance indicators, Provision of incentives for key talent, Policies to prevent talent attrition, Diversification of performance incentive models, Differentiation of executive compensation, Benchmarking against competitor, Policy design considering shareholder value, Development of a shareholder value evaluation model | Shareholder value maximization model, Design of models tailored to corporate growth stages, Alignment of performance evaluation with corporate culture, Benchmark setting and comparative analysis, Setting of individual and organizational performance ratios, Establishment of corporate performance indicators and weightings, Provision of incentives for key personnel, Diversification of performance incentive models, Differentiation of executive compensation |
| Strategy Execution | RSUs holding period setting, payment target selection criteria, Generation-based differentiation model, job-based RSUs differentiation, Grade-based differentiation model, Department-based KPI setting, Individual performance evaluation model, Short-term and long-term goal setting, long-term holding incentive, performance goal weighting setting, RSUs education program development, Employee participation incentives, Organizational awareness change initiatives | RSUs holding period setting, Payment target selection criteria, Generation-based differentiation model, Job-based RSUs differentiation, Grade-based differentiation model, Department based performance evaluation model, Individual performance evaluation model, Short-term and long-term performance evaluation model, Long-term holding incentive, RSUs education program development, Employee participation incentives |
| Evaluation & Control | Optimization of RSUs evaluation cycle, Setting of performance evaluation cycle, Reflecting performance evaluation results, providing Performance evaluation feedback, Automating performance evaluation system, Performance management feedback loop, Regular policy review, Analyzing compensation based on performance, analyzing departmental operation performance, preparing periodic reports, Establishing continuous monitoring process, Continuously improving performance indicators, Establishing performance feedback system, Conducting employee satisfaction surveys, Analyzing turnover rate, Conducting employee interviews and surveys, Regular perception surveys, Performance feedback system establishment, Strengthening the transparency of performance measurement. | Setting performance evaluation cycles, Automating performance evaluation systems, Regular policies review, Analysis of departmental operation performance, Preparing regular reports, Continuously improving performance indicators, Conducting employee satisfaction surveys, Analysis of turnover rates, Implementation of talent retention policies, Establishment of performance feedback systems, Enhancing transparency of performance measurement. |

   

based on the number of experts. For more rigorous verification, this study applied Ayre & Scally [125] threshold criteria in addition to Lawshe's criteria (Section 3.1, Table 6). The legal and accounting fields in which a total of 11 experts participated are statistically significant when the minimum CVR value is more than 0.636 (P<0.05). Since there are a total of 20 experts in the strategy, HR, and IR fields, it is statistically significant (P<0.05) when the CVR value is more than 0.5 [125].

At the end of the Delphi survey, the entire Delphi statistics were delivered to experts to provide two opportunities to maintain or modify their responses and group statistics [75,76], and the survey was terminated. Two additional revision opportunities were provided before the survey was concluded. This decision reflects prior research recommendations that typically suggest limiting repetition to 2–3 rounds as an appropriate range, considering that repeated survey requests can increase expert fatigue and reduce response rates. As a result, among all items, the 'Generation-based differentiation model' item in the Strategy·HR·IR field was ultimately excluded with a CVR of 0.30.

## 4.3. Third Delphi survey validation and final results

In the third round, participants assessed the importance of items validated in the second round using a 7-point Likert scale ranging from 1 (not important at all) to 7 (very important). After calculating the mean and standard deviation for each item, a total of 8 items with a standard deviation of 1.5 or more were excluded, as shown in 'Table 15' below. This process ensured that only items satisfying both the CVR (Content Validity Ratio) and the response consistency standard deviation criteria remained on the final list.

This study further enhanced reliability by utilizing Kendall's W. According to the interpretation criteria for Kendall's W (Section 3.1, Table 7), a value within the range of 0.3 to 0.5 indicates moderate agreement, a value within the range of 0.5 to 0.7 indicates strong agreement, and a value exceeding 0.7 indicates very strong agreement. The results table for this study, incorporating both the Delphi results and Kendall's W values, are shown in 'Table 16' and 'Table 17'. For both groups, Kendall's W values across each process ranged from a minimum of 0.43 to a maximum of 0.82. This indicates that at least moderate levels of agreement were secured for all processes, with some areas reaching levels corresponding to very strong agreement.

## 4.4. Comprehensive interpretation of Delphi results

The legal and accounting expert group showed average scores above 5 points (on a 7-point scale) for all items in the Strategy Setting phase. Notably, 'Compliance with relevant laws and accounting regulations' had the highest importance (average 6.82 points, standard deviation 0.40) and the lowest variance. 'Necessity of board resolution' and 'Tax assessment criteria' were also rated around 6 points on average, with CVRs of 0.82–1.00. This indicates that legal compliance and accounting treatment standards are perceived as the most critical factors during the initial stages of RSUs implementation.

**Table 15. Deleted items for exceeding standard deviation of 1.5.**

| Category | Item | M | SD | CVR |
|---|---|---|---|---|
| Legal/ Accounting | Applicability to unlisted companies | 4.18 | 1.89 | 0.82 |
| | Necessity of shareholders' meeting resolution | 4.82 | 1.72 | 0.64 |
| Strategy/ HR/IR | Job-based RSUs differentiation | 5.05 | 1.66 | 0.50 |
| | Grade-based differentiation model | 5.00 | 1.75 | 0.80 |
| | Long-term holding incentive | 4.20 | 1.79 | 0.50 |
| | RSUs education program development | 3.60 | 1.70 | 0.50 |
| | Employee participation incentives | 4.15 | 1.73 | 0.60 |
| | Analysis of turnover rates | 4.40 | 1.79 | 0.50 |

**Table 16. Final results of the legal/ accounting.**

| Process | Item | M | SD | CVR | Kendall's W |
|---|---|---|---|---|---|
| Strategy Setting | Compliance with relevant laws and accounting regulations | 6.82 | 0.40 | 0.82 | 0.82 |
| | Necessity of board resolution | 6.45 | 0.52 | 1.00 | |
| | Tax assessment criteria | 6.00 | 0.77 | 0.82 | |
| | Tax relief measures | 5.18 | 0.60 | 0.64 | |
| | Analysis of regulations by country | 5.27 | 0.65 | 0.64 | |
| Strategy Execution | Information disclosure policy | 6.09 | 0.54 | 0.82 | 0.48 |
| | Post-termination settlement of RSUs | 6.27 | 0.65 | 1.00 | |
| | Preparation for accounting audits and inspections | 6.64 | 0.50 | 1.00 | |
| | RSUs payment record filing | 6.36 | 0.50 | 1.00 | |
| Evaluation & Control | Internal management system operation | 5.64 | 0.81 | 1.00 | Single Item |

**Table 17. Final results of the strategy/HR/IR.**

| Process | Item | M | SD | CVR | Kendall's W |
|---|---|---|---|---|---|
| Strategy Setting | Shareholder value maximization model | 5.35 | 0.49 | 0.60 | 0.43 |
| | Design of models tailored to corporate growth stages | 5.60 | 0.68 | 0.70 | |
| | Alignment of performance evaluation with corporate culture | 5.10 | 0.55 | 0.60 | |
| | Benchmark setting and comparative analysis | 5.60 | 0.60 | 0.70 | |
| | Setting of individual and organizational performance ratios | 6.05 | 0.69 | 0.90 | |
| | Establishment of corporate performance indicators and weightings | 6.15 | 0.81 | 0.90 | |
| | Provision of incentives for key personnel | 6.45 | 0.51 | 0.90 | |
| | Diversification of performance incentive models | 6.40 | 0.50 | 0.90 | |
| | Differentiation of executive compensation | 5.50 | 0.61 | 0.60 | |
| Strategy Execution | RSUs holding period setting | 6.25 | 0.55 | 0.90 | 0.77 |
| | Payment target selection criteria | 6.60 | 0.50 | 1.00 | |
| | Department-based performance evaluation model | 6.20 | 0.62 | 0.90 | |
| | Individual performance evaluation model | 6.30 | 0.47 | 0.90 | |
| | Short-term and long-term performance evaluation model | 6.40 | 0.60 | 0.90 | |
| Evaluation & Control | Setting performance evaluation cycles | 5.90 | 0.64 | 0.80 | 0.68 |
| | Automating performance evaluation systems | 5.05 | 0.51 | 0.50 | |
| | Regular policies reviews | 6.30 | 0.47 | 0.90 | |
| | Analysis of departmental operation performance | 5.15 | 0.49 | 0.60 | |
| | Preparing regular reports | 5.00 | 0.46 | 0.60 | |
| | Continuously improving performance indicators | 6.30 | 0.57 | 0.80 | |
| | Conducting employee satisfaction surveys | 4.35 | 0.49 | 0.60 | |
| | Implementation of talent retention policies | 4.80 | 0.41 | 0.50 | |
| | Establishment of performance feedback systems | 6.40 | 0.50 | 0.90 | |
| | Enhancing transparency of performance measurement | 5.85 | 0.59 | 0.80 | |

In the Strategy Execution stage, 'Preparation for accounting audits and inspections', 'RSUs payment record filing', and 'Post-termination settlement of RSUs' also recorded high importance (around 6 points) and CVR of 1.0. This result is presumed to be due to the common perception of experts that it is essential to build a system that can respond stably to external audits and supervisory inspections after the introduction of the RSUs system. In the evaluation and control stage, the average of 5.64 points and CVR 1.0 were found in the single item 'Internal management system operation', which can

be seen as meaning that legal and accounting experts also agree on the importance of post-management of the introduction of RSUs.

Kendall's W, indicating the level of agreement, was 0.82 in the Strategy Setting stage, interpreted as 'very strong agreement' under the Delphi methodology. Conversely, Kendall's W in the Strategy Execution stage was 0.48, indicating 'moderate agreement'. This result implies that while external norms (laws, tax systems, accounting standards, etc.) to be followed when introducing RSUs are relatively clearly defined, leading to convergence in expert perceptions, the reality is that diversity exists in the actual operational systems and methods, such as beneficiary selection, holding period setting, and record-keeping/disclosure methods. Recent Delphi studies also report a tendency to view Kendall's W values above 0.3 as indicating meaningful agreement and values above 0.5 as strong agreement. This result suggests that the core items derived by the legal and accounting group have reached a sufficiently acceptable level of consensus [130].

The Strategy·HR·IR expert group showed an average score above 6 and a CVR of 0.90 for items like 'Provision of incentives for key personnel', 'Diversification of performance incentive models', and 'Establishment of corporate performance indicators and weightings' in the Strategy Setting stage. This indicates that long-term incentive systems, including RSUs, are recognized as central pillars for securing key talent and designing corporate performance indicators. 'Setting of individual and organizational performance ratios' also scored an average of 6.05 points with a CVR of 0.90, suggesting that how to allocate the weight between individual and organizational performance in RSUs design is a critical strategic issue.

In the Strategy Execution phase, 'Payment target selection criteria', 'RSUs holding period setting', 'Department-based performance evaluation model', 'Individual performance evaluation model', and 'Short-term and long-term performance evaluation model' all scored above 6 points on average and had a CVR of 0.90 or higher. This demonstrates that RSUs should be operated with the understanding that they are strategic HR and compensation tools closely integrated with the performance evaluation system, going beyond being a simple compensation mechanism.

In the evaluation and control stage, items including 'Regular policies reviews', 'Continuously improving performance indicators', 'Establishment of performance feedback systems', and 'Enhancing transparency of performance measurement' recorded relatively high average values and CVRs within the process. The average value of 'Conducting employee surveys' and 'Implementation of talent retention policies' were also lower than other items, but the CVR was 0.50 or higher, so the validity criteria were met.

The level of agreement confirmed by Kendall's W was 0.43 in the strategy establishment stage, 0.77 in the strategy execution stage, and 0.68 in the evaluation and control stage. The 0.77 level of agreement in the strategy execution stage is a very strong level of agreement, and it can be seen that experts have selected the priorities of each item with a similar perspective on the actual operation method of RSUs. 0.43 in the strategy-making phase is a moderate consensus level, which reflects that the strategy of whether to prioritize RSUs in strategic goals (growth, shareholder value, talent acquisition, etc.) is flexible, reflecting the flexibility of the strategy for contextual factors such as organizational culture, business portfolio, and industrial competition structure.

## 5. AHP results

### 5.1. AHP results of Legal/Accounting category

As mentioned in the AHP research methodology in Section 3, since commercial law, capital market law, taxation, and accounting standards to which RSUs apply operate by similar normative system within same country, the results of evaluating legal/accounting group without classifying them by industry can be checked in 'Table 18' below. In the AHP analysis result table, the local priority (L) is an eigenvector value calculated by a pairwise comparison matrix, representing the relative importance of the item within the group. The sum of all L values is 1, which determines the priority of each item.

**Table 18. AHP results for Legal/Accounting.**

| Item | L | Priority | CR |
|---|---|---|---|
| Compliance with relevant laws and accounting regulations | 0.323 | 1 | 0.084 |
| Necessity of board resolution | 0.095 | 3 | |
| Tax assessment criteria | 0.040 | 7 | |
| Tax relief measures | 0.020 | 9 | |
| Analysis of regulations by country | 0.018 | 10 | |
| Information disclosure policy | 0.063 | 4 | |
| Post-termination settlement of RSUs | 0.028 | 8 | |
| Preparation for accounting audits and inspections | 0.298 | 2 | |
| RSUs payment record filing | 0.052 | 6 | |
| Internal management system operation | 0.063 | 5 | |

The CR value, a measure of consistency, is 0.084, which is less than 0.1, maintaining a good level of consistency logically because of the pairwise comparison [90]. As for the weight, 'Compliance with relevant laws and accounting regulations' was ranked first with 0.323, showing that compliance and internal control are the most important factors to consider in the entire design and operation of RSUs system. This shows that compliance is the most important factor in terms of research [135] that the management compensation structure affects the corporate social responsibility and external reliability through the quality of internal control.

'Preparation for accounting audits and inspections' is second with a weight of 0.298, which is consistent with Abdulsalam's study [136] that showed that extensive stock-based compensation can change auditors' risk perceptions and affect audit fees and surveillance intensity. The items that ranked first and second highlight that regulatory compliance and external audit and supervision response capabilities should be key reference points in RSUs system-related law and accounting

The third-ranked factor, 'Necessity of board resolution', has a weight of 0.095. While lower than the top two items, it still holds significant importance. This reflects a tendency to recognize board resolution as an essential procedure during RSUs implementation and modification, aiming to secure the system's legitimacy and justification at the internal governance level. It demonstrates that RSUs should be treated as a governance-related matter linked to board responsibility.

Items showing medium weight are 'Information disclosure policy' (0.063, 4th) and 'Internal management system operation' (0.063, 5th). These two items demonstrate the recognition that both transparency in external disclosure and internal control/risk management systems must be established together for the stable operation of the system. While legal compliance and audit response serve as the minimum defense line, disclosure and internal management systems must function as the operational infrastructure supporting the system's long-term functionality.

Other items, including 'Post-termination settlement of RSUs' (0.028, 8th), 'Tax relief measures' (0.020, 9th), and 'Analysis of regulations by country' (0.018, 10th), showed relatively lower weights. This indicates that, for Korean companies at this point, the focus should be more on the feasibility of introducing the system, compliance with norms, and audit responses, rather than on RSUs tax optimization, differences in regulations between countries, or post-termination processing and detailed issues. 'Tax assessment criteria' (0.040, 7th) and 'RSUs payment record filing' (0.052, 6th) were also assessed as essential complementary elements for operation, but not the highest priority in terms of strategic importance.

## 5.2. AHP results for strategy/HR/IR

The purpose and intensity with which RSUs are introduced and operated from a Strategy·HR·IR perspective can vary significantly depending on industry structure and management environment. This is because demand for and expected

effects of long-term incentive systems differ based on factors such as asset intensity, economic sensitivity, workforce and technology structure, R&D proportion, and regulatory intensity. To account for industry heterogeneity, this study conducted AHP analysis for four out of five industry categories created by referencing the 'Global Industry Classification Standard' jointly developed by MSCI and S&P Dow Jones Indices, as described in 'Section 3.2. (Table 8)'. The four industry groups are: (1) Consumer Staples, Consumer Discretionary (Group 1), (2) Energy, Materials, Utilities (Group 2), (3) Industrials, Real Estate (Group 3), (4) Information Technology, Communication Services (Group 4).

To maintain consistency within this study, the reliability criterion CR was set below 0.1 [90]. This level indicates that the expert judgments within each group maintain a logically acceptable level of consistency and corresponds to a strict level, as explained in 'Section 3'. The L (local priority) values presented in the results table represent the relative importance of each item within that process. The L values within a single process were normalized so that their sum equals 1. G (global priority) represents the overall hierarchical absolute importance, calculated by multiplying the weight of the higher-level process (strategy setting/execution/evaluation and control) by the L value of each item. This enabled the presentation of results showing priority by industry group and patterns across industries.

**5.2.1. AHP results for group 1.** The AHP results for Group 1 are shown in 'Table 19' below. The CR values for each detailed process were Strategy Setting at 0.096, Strategy Execution at 0.088, and Evaluation & Control at 0.090. All values were below 0.1, confirming that all processes maintain an acceptable level of logical consistency [90]. While Group 1's three process stages show similar weightings, Strategy Execution—closely tied to actual on-site performance creation—showed a slightly higher proportion. This indicates that in the consumer goods industry, the RSUs system must be established considering who receives incentives and under what rules they are executed.

In the Strategy Setting phase, 'Diversification of performance incentive models' and 'Provision of incentives for key personnel' emerged as the top and second priorities overall, respectively, while 'Shareholder value maximization model', 'Benchmark setting and comparative analysis', and 'Differentiation of executive compensation' showed relatively lower priority.

This result suggests in the consumer goods industry, the core objective of RSUs in compensation systems is less about maximizing shareholder value or refining executive compensation levels, and more about how to design a performance-linked incentive portfolio for diverse roles and levels—such as store operations and marketing—and how to retain key talent groups over the long term. Indeed, the fact that the performance of consumer goods and retail companies is largely explained by the role behaviors and competencies of customer-facing personnel has been repeatedly confirmed in meta-analyses of human resource management and store-level empirical studies [117,137–139]. Considering this prior research, the finding that a combination of various incentive tools and key talent incentives ranked highest in Group 1 demonstrates that RSUs should be designed as one pillar of a multi-layered incentive architecture, alongside short-term rewards such as store incentives, sales commissions, and campaign bonuses.

In the Strategy Execution phase, 'RSUs holding period setting' and 'Payment target selection criteria' were jointly ranked 3rd overall and selected as the top priority in this process. Following these, 'Department-based performance evaluation model', 'Short-term and long-term performance evaluation model', and 'Individual performance evaluation model' also ranked within the top 10 overall, forming the upper tier. Here, within an industry structure characterized by significant pressure on short-term metrics like sales, inventory turnover, and promotion performance, the RSUs system demonstrates that concrete payment rules and performance evaluation methods function as key variables determining system acceptance and motivation, rather than abstract strategic slogans. Existing research on compensation by sales organizations has also revealed that compensation structure and evaluation period have a significant impact on salespeople's effort level, product mix, and customer management [140,141]. In Group 1, RSUs retention period, target selection, department/person/long-term and short-term evaluation models are at the top of the list, which shows the intention to prevent employees from focusing only on short-sighted operations due to the industry's strong short-term pressure on sales, and to use the RSUs system as a device to link responsibility and compensation for mid to long-term tasks such as long-term brand building, enhancing customer lifetime value, and digital transformation.

**Table 19. Group 1 of AHP results for strategy/HR/IR.**

| Process | Item | L | G | Priority | CR |
|---|---|---|---|---|---|
| Strategy Setting L(0.335) | Provision of incentives for key personnel | 0.267 | 0.089 | 2 | 0.096 |
| | Diversification of performance incentive models | 0.358 | 0.120 | 1 | |
| | Design of models tailored to corporate growth stages | 0.072 | 0.024 | 15 | |
| | Alignment of performance evaluation with corporate culture | 0.114 | 0.038 | 11 | |
| | Establishment of corporate performance indicators and weightings | 0.078 | 0.026 | 14 | |
| | Setting of individual and organizational performance ratios | 0.040 | 0.014 | 18 | |
| | Differentiation of executive compensation | 0.032 | 0.011 | 22 | |
| | Benchmark setting and comparative analysis | 0.020 | 0.007 | 23 | |
| | Shareholder value maximization model | 0.019 | 0.007 | 24 | |
| Strategy Execution L(0.345) | RSUs holding period setting | 0.249 | 0.086 | 3 | 0.088 |
| | Payment target selection criteria | 0.249 | 0.086 | 3 | |
| | Department-based performance evaluation model | 0.217 | 0.075 | 6 | |
| | Individual performance evaluation model | 0.125 | 0.043 | 10 | |
| | Short-term and long-term performance evaluation model | 0.161 | 0.055 | 8 | |
| Evaluation & Control L(0.321) | Setting performance evaluation cycles | 0.233 | 0.075 | 5 | 0.090 |
| | Automating performance evaluation systems | 0.164 | 0.053 | 9 | |
| | Regular policies reviews | 0.211 | 0.068 | 7 | |
| | Analysis of departmental operation performance | 0.090 | 0.029 | 12 | |
| | Preparing regular reports | 0.084 | 0.027 | 13 | |
| | Continuously improving performance indicators | 0.058 | 0.019 | 16 | |
| | Conducting employee satisfaction surveys | 0.047 | 0.015 | 17 | |
| | Implementation of talent retention policies | 0.038 | 0.012 | 20 | |
| | Establishment of performance feedback systems | 0.034 | 0.011 | 21 | |
| | Enhancing transparency of performance measurement | 0.040 | 0.013 | 19 | |

In Evaluation and Control, 'Setting performance evaluation cycles' (5th overall), 'Regular policies reviews' (7th overall) and 'Automating performance evaluation systems' (9th overall) ranked high within in the process. The counter-argument 'Conducting employee satisfaction surveys', 'Continuously improving performance indicators' and 'Establishment of performance feedback systems' showed low weights in the process. It views the priority in the business environment of consumer goods companies, which have to manage many stores, branches, and large-scale manpower, should be standardized in the performance evaluation cycle, regularly reviewed methods and systems to secure fairness and pursue efficiency with an automated evaluation system. It has been repeatedly verified in previous studies that acceptance and recognition of procedural fairness in performance management systems directly affect employees' attitudes and performance [117,142]. Recent research also highlights that transparency and explain-ability in the compensation process strengthen organizational trust and motivational climate, providing valuable insights for this process [143].

**5.2.2. AHP result of group 2.** The AHP results for Group 2 are shown in 'Table 20' below. The CR values for each detailed process were Strategy Setting 0.045, Strategy Execution 0.094 and Evaluation and Control 0.097, all below 0.1. This indicates that the pairwise comparisons made by resource and energy industry experts achieved a level of logical consistency acceptable for acceptance [90]. The weighting for the three processes assigned the highest proportion to Strategy Setting (0.370), while the remaining two processes were evaluated at nearly identical levels. This demonstrates that how the designed institutional framework operates is critically important in the resource and energy industry, given its characteristic of involving numerous long-term projects.

**Table 20. Group 2 of AHP results for strategy/HR/IR.**

| Process | Item | L | G | Priority | CR |
|---|---|---|---|---|---|
| Strategy Setting L(0.370) | Provision of incentives for key personnel | 0.216 | 0.080 | 2 | 0.045 |
| | Diversification of performance incentive models | 0.198 | 0.073 | 3 | |
| | Design of models tailored to corporate growth stages | 0.147 | 0.054 | 7 | |
| | Alignment of performance evaluation with corporate culture | 0.102 | 0.038 | 12 | |
| | Establishment of corporate performance indicators and weightings | 0.103 | 0.038 | 11 | |
| | Setting of individual and organizational performance ratios | 0.075 | 0.028 | 18 | |
| | Differentiation of executive compensation | 0.063 | 0.023 | 19 | |
| | Benchmark setting and comparative analysis | 0.054 | 0.020 | 21 | |
| | Shareholder value maximization model | 0.042 | 0.015 | 23 | |
| Strategy Execution L(0.321) | RSUs holding period setting | 0.265 | 0.085 | 1 | 0.094 |
| | Payment target selection criteria | 0.216 | 0.069 | 4 | |
| | Department-based performance evaluation model | 0.168 | 0.054 | 8 | |
| | Individual performance evaluation model | 0.188 | 0.060 | 5 | |
| | Short-term and long-term performance evaluation model | 0.164 | 0.053 | 9 | |
| Evaluation & Control L(0.321) | Setting performance evaluation cycles | 0.181 | 0.056 | 6 | 0.097 |
| | Automating performance evaluation systems | 0.105 | 0.033 | 14 | |
| | Regular policies reviews | 0.125 | 0.039 | 10 | |
| | Analysis of departmental operation performance | 0.121 | 0.037 | 13 | |
| | Preparing regular reports | 0.101 | 0.031 | 15 | |
| | Continuously improving performance indicators | 0.092 | 0.029 | 17 | |
| | Conducting employee satisfaction surveys | 0.098 | 0.030 | 16 | |
| | Implementation of talent retention policies | 0.075 | 0.023 | 20 | |
| | Establishment of performance feedback systems | 0.047 | 0.015 | 24 | |
| | Enhancing transparency of performance measurement | 0.054 | 0.017 | 22 | |

In the Strategy Setting phase, 'Provision of incentives for key personnel' and 'Diversification of performance incentive models' ranked first and second within the process, and second and third overall. Next, 'Design of models tailored to corporate growth stages' ranked seventh overall. This indicates that respondents in this industry group emphasized designing RSUs system focused on a long-term incentive portfolio targeting key talent, rather than the entire workforce. Conversely, 'Shareholder value maximization model', 'Benchmark setting and comparative analysis', and 'Differentiation of executive compensation' were rated relatively low, ranking 19th to 23rd. This is because, in the resource and energy industry, which centers on large-scale capital investment and long-term projects, the key to performance creation lies in the accumulated capabilities and collaboration of specialized workforce groups—such as engineers, field operations personnel, and project managers—rather than a few top executives. Therefore, in this industry group, designing a long-term incentive combination that reflects diverse threat factors and performance profiles, and how to stably retain key talent groups who will remain with the organization for the long term, emerges as a priority task in the strategy formulation stage. These results can be interpreted in line with meta-analysis findings that human resource management systems contribute to financial performance by mediating talent retention and organizational capability building [117].

RSUs strategies in the resource and energy industry may assume new roles amid recent pressures for eco-friendly management, driven by energy transition and strengthened ESG regulations. Energy companies hold large-scale capital investments and carbon-intensive assets. In a situation where the international transition of low-carbon and eco-friendly portfolios focusing on renewable energy is required [144,145], the long-term incentive system can be used as a means of compensation for long-term goals that induce climate risk management and expansion of energy conversion investment.

Ritz [146], an analysis of global major energy companies, showed that major companies include carbon emissions reduction, renewable energy share, safety, and environmental indicators in their performance indicators of long-term incentives. This means that compensation systems linked to climate response and sustainability are gradually spreading, and it is recommended that Group 2 companies strategically design RSUs with important weight on sustainability goals along with financial performance or regulatory compliance [147].

Most of the items in the Strategy Execution stage ranked high in the overall process, showing that the resource and energy industries needed to focus on RSUs execution. 'RSUs holding period setting' showed the highest weight as the first overall priority, followed by 'Payment target selection criterion' (4th overall). 'Individual performance evaluation model' (5th overall), 'Setting performance evaluation cycles' (6th overall), 'Department-based performance evaluation model' (8th overall), and 'Short and long performance evaluation model' (9th overall) ranked high in the entire process. As it has a project-based business in this industry and has an investment and recovery cycle of as little as years to as much as 10 years, RSUs' retention period can function as a device that binds core technologies and project personnel to the organization in the mid to long term and can also enhance organizational commitment.

Abudy and Benninga [148] noted that due to illiquidity and non-transferability, even stock compensation of the same nominal value requires employees to demand a significant risk premium relative to cash. The findings of this study suggest that when operating RSUs program over the long term, the holding period, forfeiture conditions, and risk-sharing structure must be carefully calibrated. Furthermore, recent research demonstrates that long-term equity-based incentives significantly influence management's risk preferences, innovation investments, and long-term value creation [149]. This supports the necessity of prioritizing RSUs holding periods and performance evaluation systems in capital-intensive and regulation-intensive industries.

Park et al. [4] report that in industries heavily reliant on facilities and regulations, such as resources and energy, the effects of introducing RSUs tend to be reflected gradually over the long term rather than through short-term improvements in EPS. Considering these points together, the combination of 'holding period–selection of recipients–short- and long-term performance evaluation model–evaluation cycle' derived in this study can be understood as showing how the system should be structured in the implementation phase to realize long-term performance.

Xu et al. [147] present findings from an analysis of technology, capital and labor-intensive energy firms, showing that innovative investment and executive incentives have a positive combined effect on long-term financial sustainability. Sulimany [150] similarly uses S&P 500 data to show that R&D expenditure, while potentially impairing short-term profitability, significantly improves financial sustainability after a lag period. Considering this prior research, Group 2 companies should operate RSUs not merely as a cost compensation tool but as a human resource strategy driving long-term performance.

In the Evaluation and Control stage, individual items were not highly prioritized, and the differences between them were modest. 'Setting performance evaluation cycles' received the highest weighting in the process (6th overall), followed by 'Regular policies reviews' (10th overall) and 'Analysis of departmental operation performance' (13th overall). The items of 'Implementation of talent retention policies', 'Enhancing Transparency of performance measurement' and 'Establishment of performance feedback systems' were ranked at the bottom of the 20th to 24th place entirely. This can be said to emphasize the prudence and importance of the system design stage once again in this industry, where projects are long-lived and fluctuate in economy, international raw material prices, and various regulations. This means frequent or hasty system changes and involvement should be avoided due to excessive evaluation and control, and at the same time, RSUs operation data and member responses should be regularly collected and analyzed in a fair and consistent manner at the same time as the macroeconomic situation of critical issues in a business or project. [151].

**5.2.3. AHP results for group 3.** The results of the AHP pairwise comparison of Group 3 (industry and infrastructure) are shown in 'Table 21' below. The CR value of each process are all less than 0.1, so the pairwise comparison results have logical consistency [90]. As for the weight for each process, the proportion of Strategy Execution (0.346), which is closely related to the implementation of the system, was slightly higher.

**Table 21. Group 3 of AHP results for strategy/HR/IR.**

| Process | Item | L | G | Priority | CR |
|---|---|---|---|---|---|
| Strategy Setting L(0.328) | Provision of incentives for key personnel | 0.165 | 0.054 | 8 | 0.049 |
| | Diversification of performance incentive models | 0.171 | 0.056 | 6 | |
| | Design of models tailored to corporate growth stages | 0.135 | 0.044 | 10 | |
| | Alignment of performance evaluation with corporate culture | 0.090 | 0.030 | 15 | |
| | Establishment of corporate performance indicators and weightings | 0.185 | 0.061 | 4 | |
| | Setting of individual and organizational performance ratios | 0.086 | 0.028 | 16 | |
| | Differentiation of executive compensation | 0.057 | 0.019 | 23 | |
| | Benchmark setting and comparative analysis | 0.066 | 0.022 | 21 | |
| | Shareholder value maximization model | 0.044 | 0.014 | 24 | |
| Strategy Execution L(0.346) | RSUs holding period setting | 0.247 | 0.086 | 1 | 0.090 |
| | Payment target selection criteria | 0.216 | 0.075 | 2 | |
| | Department-based performance evaluation model | 0.172 | 0.059 | 5 | |
| | Individual performance evaluation model | 0.216 | 0.075 | 2 | |
| | Short-term and long-term performance evaluation model | 0.150 | 0.052 | 9 | |
| Evaluation & Control L(0.325) | Setting performance evaluation cycles | 0.172 | 0.056 | 7 | 0.093 |
| | Automating performance evaluation systems | 0.073 | 0.024 | 20 | |
| | Regular policies reviews | 0.106 | 0.034 | 13 | |
| | Analysis of departmental operation performance | 0.060 | 0.019 | 22 | |
| | Preparing regular reports | 0.076 | 0.025 | 19 | |
| | Continuously improving performance indicators | 0.086 | 0.028 | 17 | |
| | Conducting employee satisfaction surveys | 0.123 | 0.040 | 12 | |
| | Implementation of talent retention policies | 0.128 | 0.042 | 11 | |
| | Establishment of performance feedback systems | 0.092 | 0.030 | 14 | |
| | Enhancing transparency of performance measurement | 0.083 | 0.027 | 18 | |

The highest importance item in the Strategy Setting process was 'Establishment of corporate performance indicators and weightings', ranking entire 4th (L = 0.185), followed by 'Diversification of performance incentive models' (6th overall) and 'Provision of incentives for key personnel' (8th overall). 'Shareholder value maximization model', 'Differentiation of executive compensation' and 'Benchmark setting and comparative analysis' positioned 21st to 24th overall and were evaluated as low importance based on the whole process. This indicates that companies in the industrial and infrastructure sectors should prioritize designing specific performance metric structures and incentive portfolios that reflect project-level outcomes, along with compensation structures for key technical and managerial personnel leading project execution, when designing compensation systems including RSUs.

In the Strategy Execution phase, 'RSUs holding period setting' emerged as the top priority among all items. 'Payment target selection criteria' and 'Individual performance evaluation model' tied for second place overall, followed by 'Department-based performance evaluation model' in fifth place and 'Short-term and long-term performance evaluation model' in ninth place overall. In project-based business such as plants, construction, engineering and real estate development in which these companies operate, the departure of key engineers and project managers directly leads to project delays and increased costs. That is why it is necessary to strategically design the retention period and grant target of RSUs to establish a mechanism that induces long-term retention and contribution until the project is completed. Abudy and Benninga [148] viewed RSUs, which are difficult to cash in immediately, can function as a strong structure that attracts long-term employment and performance contributions, although discounts occur against nominal value due to illiquidity and non-diversification constraints. Among the items, the RSUs retention period, payment target, individual/

department units, and long and short-term performance models all showed high importance, suggesting that in the project environment (especially large scale), the RSUs system should be designed to be flexibly adjusted according to the project progress stage and performance rather than to operate with only a fixed contract once setting.

Han et al. [152] using large-scale construction projects as a case study, propose a dynamic incentive model combining explicit incentives with reputation effects. They analyze contractors maintain high effort levels when reputation accumulation and rewards are combined throughout the project period. Liu and Zhou [153] similarly analyze cooperative innovation behavior among mega-project participants from a reward-penalty mechanism perspective, emphasizing that willingness to participate in innovation is stably maintained when the reward structure reflects both short-term outputs and long-term collaborative outcomes. These prior studies support the conclusion from Group 3 that RSUs should be utilized as a dynamic incentive mechanism to induce the retention and cooperation of key personnel throughout the project lifecycle.

In the Evaluation and Control phase, 'Setting performance evaluation cycles' showed the highest importance (7th overall), followed by 'Implementation of talent retention policies' (11th overall), 'Conducting employee satisfaction surveys' (12th overall), 'Regular policy review' (13th overall), and 'Establishment of performance feedback systems' (14th overall) showed relatively high weightings. 'Continuously improving performance indicators', 'Enhancing transparency of performance measurement', 'Preparing regular reports', 'Automating performance evaluation systems', and 'Analysis of departmental operation performance' ranked lower, but these items also serve as auxiliary mechanisms for post-hoc verification and adjustment of project outcomes and system operation results.

Considering that low fairness and predictability in evaluation and reward systems within organizations executing large-scale, long-term projects can lead to critical risks such as key personnel attrition and reduced organizational commitment, the high rankings of 'Implementing talent retention policies' and 'Conducting employee satisfaction surveys' in Group 3, and 'Regular policy review' in Group 3 being positioned at the top of the evaluation and control domain reflects a managerial intent to maintain the acceptability and perceived fairness of the performance management system. DeNisi and Smith [142] report that perceptions of fairness in performance management systems directly influence employee attitudes and performance. Tenhiälä et al. [143] also demonstrate that career outcomes for employees can vary depending on which behaviors are rewarded by performance management and compensation practices. This suggests that in Group 3, the evaluation and control processes may enhance their effectiveness when combined with the long-term performance stability and workforce retention mechanisms of the RSUs system.

**5.2.4. AHP results for group 4.** The AHP Survey results of Group 4 (technology, communication industry) are 'Table 22' below. The CR values for each process are all less than 0.1, and the results of the expert pairwise comparison have logical consistency [90]. The weights of the three processes were 0.370 for Strategy Setting, 0.321 for Strategy Execution, and 0.309 for Evaluation and Control, which were not significant differences, but the Strategy Setting step accounted for the highest proportion.

In the Strategy Setting stage, 'Provision of incentives for key personnel' is the first overall priority and 'Diversification of performance incentive models' is the second entire priority. 'Design of models tailored to corporate growth stages' is the 6th overall ranking, 'Alignment of performance evaluation with corporate culture', 'Shareholder value maximization model', 'Benchmark setting and comparative analysis' and 'Differentiation of executive compensation' are ranked 19–24 overall. From this point of view, Group 4 companies recognize RSUs as a means of securing and maintaining strategic core talents including key technical personnel, IT system developers, product/service managers and AI/data experts, and prioritize system design. Prakash et al. [154] highlight the importance of competitive compensation, ability recognition, work environment, job autonomy, and long-term incentive packages as key factors that lower turnover intentions for the IT industry.

The fact that HR and strategy managers in the technology and telecommunications industries prioritize 'core talent incentives' and 'diverse compensation portfolios' over refining shareholder value models during the strategic planning phase aligns precisely with this context. This strongly reflects the perception that RSUs are structural mechanisms

**Table 22. Group 4 of AHP results for strategy/HR/IR.**

| Process | Item | L | G | Priority | CR |
|---|---|---|---|---|---|
| Strategy Setting L(0.370) | Provision of incentives for key personnel | 0.260 | 0.096 | 1 | 0.041 |
| | Diversification of performance incentive models | 0.205 | 0.076 | 2 | |
| | Design of models tailored to corporate growth stages | 0.159 | 0.059 | 6 | |
| | Alignment of performance evaluation with corporate culture | 0.113 | 0.042 | 11 | |
| | Establishment of corporate performance indicators and weightings | 0.087 | 0.032 | 16 | |
| | Setting of individual and organizational performance ratios | 0.060 | 0.022 | 18 | |
| | Differentiation of executive compensation | 0.052 | 0.019 | 19 | |
| | Benchmark setting and comparative analysis | 0.036 | 0.013 | 23 | |
| | Shareholder value maximization model | 0.027 | 0.010 | 24 | |
| Strategy Execution L(0.321) | RSUs holding period setting | 0.226 | 0.073 | 3 | 0.082 |
| | Payment target selection criteria | 0.226 | 0.073 | 3 | |
| | Department-based performance evaluation model | 0.226 | 0.073 | 3 | |
| | Individual performance evaluation model | 0.150 | 0.048 | 8 | |
| | Short-term and long-term performance evaluation model | 0.172 | 0.055 | 7 | |
| Evaluation & Control L(0.309) | Setting performance evaluation cycles | 0.114 | 0.035 | 14 | 0.083 |
| | Automating performance evaluation systems | 0.131 | 0.040 | 12 | |
| | Regular policies reviews | 0.114 | 0.035 | 14 | |
| | Analysis of departmental operation performance | 0.151 | 0.047 | 9 | |
| | Preparing regular reports | 0.140 | 0.043 | 10 | |
| | Continuously improving performance indicators | 0.117 | 0.036 | 13 | |
| | Conducting employee satisfaction surveys | 0.073 | 0.023 | 17 | |
| | Implementation of talent retention policies | 0.062 | 0.019 | 20 | |
| | Establishment of performance feedback systems | 0.048 | 0.015 | 22 | |
| | Enhancing transparency of performance measurement | 0.049 | 0.015 | 21 | |

designed to retain core talent groups within the organization long-term, rather than merely supplementary tools to gain an edge in salary competition. In this regard, Liu et al. [155] statistically proved that companies implementing restricted stock-like schemes show a significantly lower turnover rate among key personnel like executives. Their study verified whether equity-based incentive systems for executives at Chinese listed companies actually function as 'golden handcuffs'.

In the Strategy Execution stage, 'RSU holding period setting', 'payment target selection criteria', and 'Department-based performance evaluation model' all ranked 3rd overall, forming the top group for this process. Following this, 'Short-term and long-term performance evaluation model' ranked 7th overall, and 'Individual performance evaluation model' ranked 8th overall.

Products and services in the technology and telecommunications industries often have short lifecycles and performance measured on a project basis. In this environment, RSUs take on the nature of a performance contract directly linked to project success, rather than merely being a supplement to base salary. If the holding period is too short, the step-based vesting structure may not sufficiently fulfill its talent retention and engagement function. Conversely, if it is excessively long, it could reduce workforce flexibility in the high-risk, high-volatility technology industry. Yi and Oh [156] found that while CEO restricted stock like RSUs may constrain short-term R&D spending, they tend to increase innovation outputs measured by patents and citations. This suggests RSUs can serve as a mechanism to drive visible innovation outcomes at the project and product level in a compressed timeframe. Furthermore, the fact that both departmental and individual performance evaluation models, as well as short-term and long-term performance evaluation models, rank highly in Group 4 indicates that technology and telecommunications companies should consider adopting performance

management strategies that simultaneously capture both short-term sprint/quarterly performance and long-term delayed outcomes like platform building and network effects.

In the Evaluation and Control stage, while priority gaps between items are relatively moderate, distinctions in importance can be made. 'Analysis of departmental operation performance' ranks 9th overall, 'Preparing regular reports' ranks 10th, 'Continuously improving performance indicators' ranks 13th, and 'Automating performance evaluation systems' ranks 12th, all showing relatively high weightings. In contrast, 'Conducting employee satisfaction surveys' ranks 17th overall, 'Implementing talent retention policies' ranks 20th overall, 'Establishing performance feedback systems' ranks 22nd overall, and 'Enhancing transparency of performance measurement' ranks 21st overall, remaining in lower positions. This indicates that for Group 4 companies, rather than designing new systems or policies at the evaluation and control stage, it is more important to monitor and fine-tune how the already designed RSUs and performance management systems are functioning in the field based on data.

**5.2.5. Strategy/HR/IR domain AHP results synthesis and discussion.** Below 'Table 23' synthesizes the item priority rankings from the AHP results across the four industry groups (Group 1–4). In this table, G stands for Group. One interesting point is that the rankings for each item vary significantly across industries. This supports the research question of this study: rather than introducing and operating RSUs as a 'standard package' all at once, a separate design framework tailored to each industry group is necessary.

Table 23. Priority comparison of industry priorities in strategy/HR/IR.

| Process | Item | G1 | G2 | G3 | G4 |
|---|---|---|---|---|---|
| Strategy Setting | Provision of incentives for key personnel | 2 | 2 | 8 | 1 |
| | Diversification of performance incentive models | 1 | 3 | 6 | 2 |
| | Design of models tailored to corporate growth stages | 15 | 7 | 10 | 6 |
| | Alignment of performance evaluation with corporate culture | 11 | 12 | 15 | 11 |
| | Establishment of corporate performance indicators and weightings | 14 | 11 | 4 | 16 |
| | Setting of individual and organizational performance ratios | 18 | 18 | 16 | 18 |
| | Differentiation of executive compensation | 22 | 19 | 23 | 19 |
| | Benchmark setting and comparative analysis | 23 | 21 | 21 | 23 |
| | Shareholder value maximization model | 24 | 23 | 24 | 24 |
| Strategy Execution | RSUs holding period setting | 3 | 1 | 1 | 3 |
| | Payment target selection criteria | 3 | 4 | 2 | 3 |
| | Department-based performance evaluation model | 6 | 8 | 5 | 3 |
| | Individual performance evaluation model | 10 | 5 | 2 | 8 |
| | Short-term and long-term performance evaluation model | 8 | 9 | 9 | 7 |
| Evaluation & Control | Setting performance evaluation cycles | 5 | 6 | 7 | 14 |
| | Automating performance evaluation systems | 9 | 14 | 20 | 12 |
| | Regular policies reviews | 7 | 10 | 13 | 14 |
| | Analysis of departmental operation performance | 12 | 13 | 22 | 9 |
| | Preparing regular reports | 13 | 15 | 19 | 10 |
| | Continuously improving performance indicators | 16 | 17 | 17 | 13 |
| | Conducting employee satisfaction surveys | 17 | 16 | 12 | 17 |
| | Implementation of talent retention policies | 20 | 20 | 11 | 20 |
| | Establishment of performance feedback systems | 21 | 24 | 14 | 22 |
| | Enhancing transparency of performance measurement | 19 | 22 | 18 | 21 |

Consumer goods companies (Group 1) show strong interest in the structure and breadth of their incentive portfolio from the strategy formulation stage. They prioritize determining the role RSUs will play within a multi-layered performance incentive system and which job/grade groups will be the core targets for long-term incentives. During the strategy execution phase, the focus shifts to how to differentially allocate RSUs across various customer-facing roles like stores, sales, and marketing. In the evaluation and control phase, the practical feasibility of evaluation cycles and system-based operations is relatively emphasized to manage a large number of employees stably. In other words, the RSUs framework for the consumer goods industry must be designed with a primary focus on combining it with existing short-term incentives, its applicability to large fieldwork forces, and managing operational complexity.

Resource and energy companies (Group 2) make different choices. Given that performance indicators and weight structures are high in the strategy setting stage, and the combination of individual, department, short and long-term evaluation models is evaluated as high importance in the strategy execution stage, companies in this industry should focus on multidimensional indicators and weight systems that can reflect individual performance, project division performance and organizational level performance in the RSUs system.

Comparing the results of the evaluation and control stages with other industries, devices that maintain trust in human resource leakage prevention policies, employee experience monitoring, and institutional fairness have relatively high priorities. Therefore, the RSUs framework for the resource and energy industry should focus structurally on designing how long to grant, with which risks to combine, and to which project stages to link—rather than solely on how much to grant.

Industrial and infrastructure companies (Group 3) exhibit a different priority configuration. When introducing RSUs, they relatively prioritize multidimensional performance metrics and weighting systems that can reflect not only specific individuals but also project/sector and organizational-level performance. In the strategy formulation stage, performance metrics and weighting structures rank highly, and in the strategy execution stage, the combination of individual, departmental, and short-term/long-term evaluation models is also assessed as highly important. Simultaneously, during the evaluation and control phase, mechanisms to prevent talent drain, monitor employee experience, and maintain trust in the system's fairness rank relatively higher compared to other industries. This can be interpreted as reflecting an intent to design RSUs not merely as a wage supplement, but as a tool to simultaneously manage project completion and retain key personnel in industries where project delays or quality risks are critical, such as large-scale plants, construction, transportation and logistics, and real estate development. The RSUs framework for this sector should incorporate a project KPI structure, weighting, and fairness management mechanisms alongside holding periods and recipient selection.

Technology and telecommunications companies (Group 4) place the highest weight on the strategy setting stage and the importance of key talent incentive items among the four groups. Given their environment of rapid technology cycles and talent competition, companies in this sector strongly perceive RSUs as a 'precise tool for retaining key talent'. At the strategy formulation stage, key talent incentives, diversification of incentive models, and eligibility criteria rank highest. At the strategy execution stage, the combination of individual performance evaluation and short/long-term performance models is emphasized. In the evaluation and control phase, rather than designing entirely new systems, agility—the ability to frequently adjust and update detailed parameters of RSUs design through data-driven monitoring and automated systems—emerges as a relatively critical issue. Consequently, the RSUs framework in the technology and telecommunications industries must be designed and operated based on the principles of 'targeting key talent, linking to individual performance, and data-driven fine-tuning'.

This demonstrates that even when evaluating the same 24 items, the differing priorities of processes and items across industry sectors indicate that the criteria for designing RSUs systems are strongly defined not by general corporate characteristics (size, listing status, etc.), but by industry-specific business models, project structures, and talent market dynamics. Recent compensation research also warns that a tendency toward a 'one-size-fits-all' structure may actually undermine corporate value [157]. In ESG and executive compensation research, heterogeneity analyses repeatedly report

that identical incentive structures do not yield the same effects across all company types; the direction and magnitude of effects vary depending on company type and industry [158,159].

## 6. Conclusion

Unlike previous studies [4,10,42] that conducted post-implementation quantitative research on RSUs in countries where the system had already reached maturity or in companies that had already adopted it, this study focused on the prospective question of how RSUs should be designed and operated in terms of structure and process. To achieve this, the strategic management process of strategy formulation–execution–evaluation and control was established as the analytical framework. A combined Delphi–AHP method was applied to legal/accounting and strategic/HR/IR senior-level experts to systematically organize the detailed items and their relative importance required for RSUs implementation and operation by industry sector. The results confirmed differing priorities for processes and detailed items across four industries: consumer goods, resources and energy; industry and infrastructure; technology and communications.

The academic significance of this study lies in shifting the focus from treating RSUs merely as outcome variables affecting performance to examining which design elements should be prioritized in specific contexts. Previous empirical studies have estimated average relationships between contractual variables such as RSU adoption, vesting periods, grant sizes and financial performance [9,11,21]. However, this study is significant in that it decomposes the introduction and operation of RSUs from a process perspective, premised on Korea's institutional and industrial environment, and compares and structures which management items are key at each stage by industry, thereby organizing an industry-specific application framework for RSUs.

This study also contributes methodologically by applying the Delphi–AHP methodology to the field of compensation system design. Until now, Delphi–AHP has primarily been utilized in public policy, infrastructure, quality management, and ESG compensation indicator selection [107,109]. It specifically presents a procedure for quantifying items refined through expert consensus by applying the Delphi–AHP methodology to complex compensation system research like RSUs, where law, finance, strategy, HR, and IR are simultaneously intertwined. Notably, conducting the Delphi separately for legal/accounting experts and strategy/HR/IR experts, then applying AHP to the resulting outcomes, demonstrates a model for sequentially utilizing multiple expert groups in future compensation system and governance research, serving as a benchmark study. And this study empirically supports that compensation systems such as RSUs inevitably have different configurations for each industry and demonstrated that sophisticated design utilizing the context and characteristics of the industry is needed when conducting follow-up studies to analyze the relationship between compensation systems and performance in the future.

This research also provides a variety of practical implications. First, the RSUs implementation, operation, and management items and the industry-specific weights by process derived in this study can serve as a checklist and roadmap for Korean companies considering RSUs adoption. At the strategy formulation stage, companies can check which performance metrics and weighting systems to prioritize first. At the strategy execution stage, they can assess what infrastructure to build in terms of HR, organization, and communication. In the evaluation and control stage, it is possible to systematically check the improvements of the RSUs system by item through post-evaluation and monitoring. It can also contribute to reducing trial and error by providing approaches and priority information for the introduction, operation, and management of the RSUs system to companies that recognize the need for a long-term incentive system. This can also be used as a reference model for companies and institutions in countries with a similar business environment to Korea in terms of governance, capital market system, and industrial structure. From the perspective of supervisors and legislators, discussions on laws and systems related to RSUs are not just a mere transplantation of United States and Europe, but rather to understand what risk factors and design factors are recognized as important by industry expert groups.

In addition, this study was designed on the premise of a Korean (continental legal system) institutional environment, but it also provides some implications when compared to common-law countries [160,161]. In the Anglo-American legal

world, stock-based compensation is widely used in an environment where shareholder rights protection and disclosure and contracting infrastructures are relatively strong, and the focus of contract design tends to shift toward the elaboration of performance–compensation links and the refinement of stock-based incentive design (performance standards, vesting, and incentive structures) rather than the "justification for adoption" [162,163].

On the other hand, in the early stages of introduction, such as in Korea, uncertainty management in legal and accounting interpretations and stakeholder persuasion to ensure institutional consistency can act as relatively more important prerequisites [160,161]. Therefore, the process-based (establishment–execution–evaluation/control) priority framework presented by this study can be used as a common analytical framework to compare and diagnose how the core levers of performance-indicator design and incentive risk–reward structures (e.g., performance-conditional vesting, risk sensitivity, etc.) move depending on institutional maturity (initial–growth–maturity), even if the legal system is different [162,164,165]. Of course, this study has limitations. First, as it is a study from the perspective of managers or those establishing and operating systems, the expert group conducting the analysis was primarily composed of senior managers from large listed companies. Consequently, the perspectives of other stakeholders, such as small companies, startups, labor unions and employee representatives, and institutional investors, were not reflected. Future research should expand the panel composition to compare the priorities of diverse stakeholders or attempt cross-analysis of AHP results between expert groups.

Furthermore, this study is based on the Korean context, and generalizing its findings to countries with different institutional and regulatory environments and compensation practices may be constrained. And the sample composition did not include the financial services or healthcare industries, necessitating additional research on the RSUs framework within these sectors. Next, due to the nature of the Delphi–AHP methodology, this study focused on structuring the relative importance and priority of RSUs design and operational items. Thus, the causal effect on financial performance or shareholder value needs to be additionally confirmed through subsequent quantitative studies. These points represent limitations of this study. However, they also hold significance as they suggest future research directions: expanding the national and industrial scope and combining Delphi–AHP results with panel data analysis.

Furthermore, the Delphi-AHP analysis requires additional research to re-validate the relative importance perceived by experts at a specific point in time as it changes over time and with environmental shifts. As time passes and Korea moves beyond the initial adoption phase of the RSUs system into a growth or maturity phase, factors like regulatory risk and stakeholder persuasion, which were critical early on, may become less important over time. Instead, the routine operation and internalization of performance evaluation and compensation structures could become more significant. To capture this dynamism, future research could analyze 'Shifting priorities across the RSUs life-cycle' by integrating with models like System Dynamics.

## Supporting information

**S1 File. Delphi survey dataset.** This Excel file contains the anonymized Delphi survey responses used to identify and refine the key items for RSUs adoption and governance design. It includes item-level responses across Delphi rounds and the final consolidated item list used for the subsequent AHP stage.
(XLSX)

**S2 File. AHP pairwise comparison dataset.** This Excel file contains the AHP pairwise comparison responses and the derived priority weights used in the multi-criteria decision-making analysis. It includes the pairwise matrices (by domain/industry group, where applicable) and the computed relative importance scores used to produce the AHP results.
(XLSX)

## Author contributions

**Conceptualization:** Won Albert Park, Cheong-Yeul Park.

**Data curation:** Won Albert Park.

**Formal analysis:** Won Albert Park.

**Funding acquisition:** Won Albert Park, Cheong-Yeul Park.

**Investigation:** Won Albert Park.

**Methodology:** Won Albert Park, Elena Sernova.

**Project administration:** Won Albert Park.

**Resources:** Won Albert Park.

**Software:** Won Albert Park, Cheong-Yeul Park.

**Supervision:** Elena Sernova, Cheong-Yeul Park.

**Validation:** Cheong-Yeul Park.

**Visualization:** Won Albert Park.

**Writing – original draft:** Won Albert Park.

**Writing – review & editing:** Elena Sernova, Cheong-Yeul Park.

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
