## [Decision Letter · Decision Letter 0]

26 Nov 2025

Dear Dr. Park,

Thank you for submitting your manuscript to PLOS ONE. After careful consideration, we feel that it has merit but does not fully meet PLOS ONE’s publication criteria as it currently stands. Therefore, we invite you to submit a revised version of the manuscript that addresses the points raised during the review process.

**Key Points of Concern:**

The abstract offers conceptual context but is devoid of critical analytical (data) evidence to substantiate the assertions. Also, limitations like small sample sizes, possible expert bias, and how well the sample represents the sector should be noted.The manuscript has remarkably higher AI usage index, which could be a problem with ethics and may not be within PLOS ONE's permitted range. The authors need to make a lot of changes to the manuscript so that less than 20% of it is flagged by AI and the tone is more scholarly.While the manuscript reviews RSU governance practices extensively, it does not clearly differentiate:

-What this study contributes to current RSU/equity compensation frameworks

-How the Delphi-AHP hybrid methodology offers innovative perspectives

-What actionable implications the results have for cross-industry RSU design

Much of the discussion summarizes prior literature rather than synthesizing new contributions identified through the expert-driven AHP approach and Delphi consensus.Introduction:

-Reviewer 1 correctly notes the need for methodological grounding and updated citations.

-The last paragraph of the Introduction should clearly show how the manuscript is organized and how the subsequent sections are planned.

Literature Review:

-The review must begin with a rationale paragraph framing why a literature review on RSU governance is necessary and proceed with clearer, thematically grouped sources.

-The current review cites several seminal works but lacks recent evidence from: Behavioral finance on executive incentives, corporate governance reforms post-2020, cross-industry RSU adoption trends etc.

Add a few recent (preferably 2021 to 2025) and credible sources in citations and references. References should follow APA format and guidelines and also should be arranged in alphabetical order. Further, include the doi link to the resources as much as possible. At least 80% of references must include DOIs where available.The manuscript presents both Delphi and AHP methodologies reflecting statistical rigor. Authors should describe the population of specialists, Justify the sample size (n), Explain criteria for picking experts across industries, Clarify whether experts were evenly distributed across sectorsThe manuscript reports two rounds but does not describe: How consensus was measured (e.g., percentage, stability index), How conflicting responses were handled, and the rationale for ending after two roundsGive rationale behind “AHP targets by particular industry group”. Also, clarify how to handle missing or incomplete expert responsesThe results section shows weighted RSU governance criteria by industry, but interpretation is limited, making it hard to understand. Also, tables and figures should follow the journal’s guideline.References: should follow APA format and guidelines.  Pay close attention to the authors cited in the text, as well as the spelling, year of publication, and author verification throughout the text.

The manuscript shows potential, especially in how it uses Delphi and AHP from different fields to create industry-specific RSU governance structures. But there are some very important issues that need to be addressed, and align with journal standards.

We look forward to receiving your revised manuscript.

Kind regards,

Dipendra Karki, Ph.D.

Academic Editor

PLOS ONE

“If this manuscript is accepted for publication, the authors will receive a research grant of 5,000,000 KRW from Seoul Business School (aSSIST University) to support further academic work.”

4. Thank you for stating the following in the Funding Section of your manuscript:

“This research was supported by Seoul Business School (aSSIST University).”

“If this manuscript is accepted for publication, the authors will receive a research grant of 5,000,000 KRW from Seoul Business School (aSSIST University) to support further academic work.”

Additional Editor Comments:

Dear Author/s

Thank you for your manuscript to PLOS ONE. After review of the manuscript, the reviewers’ reports, and an internal editorial assessment, we find that the manuscript addresses a relevant and underexplored topic: the design of industry-specific RSU governance frameworks using a combined Delphi-AHP approach. This manuscript demonstrates meaningful potential but several substantial revisions are required before the manuscript can be considered for publication. Please find detailed, constructive comments as follows:

• The abstract offers conceptual context but is devoid of critical analytical (data) evidence to substantiate the assertions. Also, limitations like small sample sizes, possible expert bias, and how well the sample represents the sector should be noted.

• The manuscript has remarkably higher AI usage index, which could be a problem with ethics and may not be within PLOS ONE's permitted range. The authors need to make a lot of changes to the manuscript so that less than 20% of it is flagged by AI and the tone is more scholarly.

• While the manuscript reviews RSU governance practices extensively, it does not clearly differentiate:

-What this study contributes to current RSU/equity compensation frameworks

-How the Delphi-AHP hybrid methodology offers innovative perspectives

-What actionable implications the results have for cross-industry RSU design

• Much of the discussion summarizes prior literature rather than synthesizing new contributions identified through the expert-driven AHP approach and Delphi consensus.

•Introduction:

-Reviewer 1 correctly notes the need for methodological grounding and updated citations.

-The last paragraph of the Introduction should clearly show how the manuscript is organized and how the subsequent sections are planned.

•Literature Review:

-The review must begin with a rationale paragraph framing why a literature review on RSU governance is necessary and proceed with clearer, thematically grouped sources.

-The current review cites several seminal works but lacks recent evidence from: Behavioral finance on executive incentives, corporate governance reforms post-2020, cross-industry RSU adoption trends etc.

•Add a few recent (preferably 2021 to 2025) and credible sources in citations and references. References should follow APA format and guidelines and also should be arranged in alphabetical order. Further, include the doi link to the resources as much as possible. At least 80% of references must include DOIs where available.

•The manuscript presents both Delphi and AHP methodologies reflecting statistical rigor. Authors should describe the population of specialists, Justify the sample size (n), Explain criteria for picking experts across industries, Clarify whether experts were evenly distributed across sectors

•The manuscript reports two rounds but does not describe: How consensus was measured (e.g., percentage, stability index), How conflicting responses were handled, and the rationale for ending after two rounds

•Give rationale behind “AHP targets by particular industry group”. Also, clarify how to handle missing or incomplete expert responses

•The results section shows weighted RSU governance criteria by industry, but interpretation is limited, making it hard to understand. Also, tables and figures should follow the journal’s guideline.

•References: should follow APA format and guidelines. Pay close attention to the authors cited in the text, as well as the spelling, year of publication, and author verification throughout the text.

-The manuscript shows potential, especially in how it uses Delphi and AHP from different fields to create industry-specific RSU governance structures. But there are some very important issues that need to be addressed, and align with journal standards.

Editorial Decision: Major Revision

Looking forward to receiving a revised version that incorporates these suggestions.

Warm regards,

Academic Editor

PLOS ONE

Reviewers' comments:

Reviewer's Responses to Questions

**Comments to the Author**

1. Is the manuscript technically sound, and do the data support the conclusions?

Reviewer #1: Yes

Reviewer #2: Yes

2. Has the statistical analysis been performed appropriately and rigorously?

Reviewer #1: I Don't Know

Reviewer #2: Yes

3. Have the authors made all data underlying the findings in their manuscript fully available?

Reviewer #1: Yes

Reviewer #2: Yes

4. Is the manuscript presented in an intelligible fashion and written in standard English?

Reviewer #1: Yes

Reviewer #2: Yes

Reviewer #1: The article is well-written, with clarity in its application aspects and the relevance of the topic. It needs to better define the methodological aspects in order to better characterize the study and better inform the reader. More comments were described in the supplementary file.

Reviewer #2: The paper discusses an interesting topic and presents the main concepts, the methodology, and the findings in a thorough and coherent manner. The research has the potential to make a meaningful contribution to the field.

**Do you want your identity to be public for this peer review?** For information about this choice, including consent withdrawal, please see our Privacy Policy

Reviewer #1: No

Reviewer #2: No

---

## [Author Response · Author response to Decision Letter 1]

9 Jan 2026

<Regarding basic forms and administrative procedures>

1. (Request) Require to meet the basic format of the manuscript and PLOS ONE criteria related to the AI utilization index

(Action) We did our best to revise the basic format of the manuscript (apply font, font size, Vancouver citation style, etc.) as guided on the PLOS ONE website. In addition, we confirmed that the AI utilization index and plagiarism index are below the standard values as a result of Turnitin inspection. In addition, we added and supplemented the DOI to the reference. The percentage of data with DOI and some data without DOI (such as book and conference etc.) were supplemented by adding accessible links.

2. (Request) Provide details on participant consent

(Action) In the online submission form, I wrote down the description, method, and inclusion of minors (this study is not included), and I revised the text to ensure that the content is properly included in the research methodology section.

3. (Request) Regarding research funding

(Action) Regarding the research funding, we will delete the part that is mentioned in the text in accordance with the PLOS ONE form and submit the revised cover letter including the relevant information in the cover letter as you guided us. There were some revisions made to reflect the revisions made by reviewers through Peer Group Review, so we also submitted the revised cover letter with the revision.

<Reviewer Complement Reflection Related>

1. Abstract Reviewer Request Complementation

1-1. Request reviewer

• Abstract presents a conceptual background, but lacks key analysis and data to support the arguments.

• Limitations on samples, representativeness, and bias, such as small number of samples, possibility of expert bias, and whether the samples represent the industry, should be specified in Abstract.

• The limitation that Delphi–AHP focuses on structuring priorities and cannot directly verify the causal effect of RSU on actual financial performance and shareholder value needs to be included in Abstract.

1-2. Author Supplements

• Abstract specifies the sample composition and procedures of the 3rd Delphi (31 experts in total) and AHP panel (48 leaders in total, including industry-specific leaders) and statistically validated content validity and consensus levels with CVR and Kendall's W. It also strengthened the analytical foundation of the results, including the process of calculating priorities on a CR<0.1 basis.

• Specifying in Abstract that the sample is specific to a specific industry range, it clarified the generalization limitations of not including the financial and healthcare industries, and suggested the need for further research. (Representational/bias issues directly mentioned in Abstract in the form of 'Industrial Scope Restriction')

• Delphi–AHP stated in Abstract the limitation that, while valid for prioritization, the causal effect of RSU on financial performance and shareholder value cannot be directly verified, and that this result needs to be extended to subsequent quantitative verification studies.

2. Methodology Reviewer Request Complementation

2-1. Request reviewer

• Methodological grounding and up-to-date citations need to be reinforced

• In the final paragraph of the introduction, you must clearly state what structure the paper is structured and how each section develops thereafter.

2-2. Author Supplements

• The introduction was reconstructed as "Structural limitations of stock options → institutional and economic mechanisms of RSU (besting/no events/relaxing short-termism) → National diffusion path (US, Europe, Japan, and Korea) → Previous quantitative studies (performance, governance, HRM perspective) → Research gap (lack of design and operational decision-making framework in early-introducing countries) → Justification of Delphi–AHP approach in this study."

• In particular, to compensate for the somewhat descriptive aspects of the existing introduction, the evidence was strengthened by citing additional quantitative and empirical-based prior studies such as performance changes before and after introduction, PSM-based comparison, the relationship between contract design variables (vesting/scale, etc.), predictive power of cost information, governance interaction, and HRM mediating effects. Existing Introduction utilized about 10 citations, but netted 16 citing 26 centered on the latest paper after the modification.

• In the final paragraph of the introduction, "Composition of Chapter 2–6 and its role in each chapter (theoretical background → methodology/sample/design → Delphi/AHP results → implications and limitations)" were added to the paper roadmap to help the reader understand the development at a glance.

3. Literature Research Reviewer Request Supplement

3-1. Request reviewer

• Literature review should first begin with a logical paragraph explaining "why a literature review of RSU governance is necessary."

• Behavioral Finance Study on Management Incentives

• A Study on Corporate Governance Reform after 2020

• Cross-industry RSU adoption trends

3-2. Author Supplements

• Added a rationale paragraph that begins with "Why RSU Governance Literature Review is required" – a sense of the problem that RSU should be designed and operated in the 'institutional, governance, and industry context' in the rush to review literature, and a research gap suggesting that existing research focuses on ex post verification and lacks governance/operational framework research in the initial stage of introduction (introduction to 2.1 and connection to 2.5).

• Reinforced research on behavioral finance related to management incentives. To complement the existing "RSU effect" centered narrative, an independent section (2.2) has been established, and behavioral finance-based arguments have been systematically added, including incentives to maximize EPS, differences in risk preferences based on regulatory focus, subjective valuation of employees (liquidity and variance constraints), and earnings management/financial reporting behavior. The new citation reinforcement adds nine [34]–[42]-focused articles to the behavioral finance part.

• Since 2020, research on corporate governance reform has been reinforced. An independent section (2.3) dealing with governance reform-compensation system linkage was newly established, and the impact of recent system changes on long-term incentive structures, such as Say-on-pay, Shareholder Rights Directive II, strengthening compensation committees, disclosures and shareholder voting, and spreading stewardship codes, was summarized based on previous studies. The new citation reinforcement added six [43]–[48]-centered articles to the Governance Reform Part (including discussions on the Korean context).

• We have also strengthened the trend of introducing RSUs across industries. An independent section (2.4) was newly established and the trend of adopting RSU/stock options based on S&P corporate compensation data (annual rate) was presented in Table 1 to quantitatively clarify the "spreading trend." In addition, we systematically organized industry heterogeneity based on differences in human capital dependence/regulatory strength by industry and linked empirical results that vary in significance by industry. At least four new citations were added, including [50]–[53].

4. Methodology Reviewer Request Complementation

4-1. Request reviewer

• Statistical rigor is demonstrated using both Delphi and AHP, but the following needs to be described in more detail (definitions and characteristics of expert groups, justification for sample size, criteria for selecting experts by industry, whether experts are distributed in a balanced manner)

• We reported that we performed two Delphi rounds, but how we measured consensus (consensus ratio, stability index, etc.) and how we handled conflicting responses and the rationale for ending the study in the second round is not fully explained.

• The background logic of "Setting AHP Targets by Industry" should be presented, and how some expert responses were handled if they were missing or incomplete should also be specified.

4-2. Author Supplements

• For defining and characteristics of expert groups, justification of sample size, criteria for selection by industry, and balanced distribution, the expert groups were clearly defined into two functional groups. (1) Legal/Accounting = a group that performs gatekeeper functions such as regulation, disclosure, and IFRS, and (2) Strategy/HR/IR = a group that connects RSUs with strategic execution, talent retention, and capital market communication.

• Delphi panel size (31 people in total; 11+20) and qualifications (minimum 15 years of experience, C-level/executive/leadership level, certificate retention, etc.) are specified in the table (Table 4–5) and sample size validity is specified based on previous studies (Delphi panel recommendation scope).

• The AHP divided GICS-based industries (1–4), selected 12 (48 people in total), and presented in a table (Table 9–12) that the HR/Strategy/IR functions were designed to be balanced within each industry (explicitly demonstrating "balanced allocation").

• The reviewer said it was 2 Delphi rounds, but we did a total of 3 rounds. For the unclear issue of Delphi rounds (measuring consensus, handling conflicting responses, and grounds for termination), we have clearly rewritten Delphi into 3 rounds. ① 1st: Open-ended item collection and integration/refining, 2nd: Evaluating the 'essential' of the item and verifying the content validity with the Content Validity Ratio (including baseline: Lawshe + Ayre & Scally correction), 3rd: After the importance (7 point Likert) assessment, specify the criteria for handling large conflicting responses/variance items (excluding standard deviation ≥ 1.5 items).

• "consensus" was added for quantitative verification with Kendall's W (presenting a table of ranges from 0 to 1 and interpretation sections), and the results were designed to be statistical indicators rather than "researcher impressions". "Termination basis" was clarified to be a structure that terminated when the triple standard of (1) CVR passing + (2) meeting variance criteria + (3) confirmation of consensus with Kendall's W was met (takes into account fatigue from unnecessary repetition).

• It specified the logic of setting AHP targets by industry and handling missing and incomplete responses. The separation of AHP by industry is based on the logic that "RSU purpose, operating environment, and performance expectations vary depending on industrial characteristics (capital intensity, R&D share, regulatory intensity, etc.)", and reinforced with GICS classification + Korea Listed Distribution (representativeness figures) (four industries represent the absolute majority of KOSPI/KOSDAQ).

• The missing/incomplete response (including matrix contradiction) case specified a procedure for a secondary confirmation (re-response request) by phone, integrated, and specified that the CR ≤ 0.10 criterion was strictly applied (after mentioning mitigation criteria of some studies (0.12 to 0.20), this study conservatively applied 0.10).

• In contrast to the existing (initial methods), the revised version adds significant additional methodology-based citations to support expert selection logic (gatekeeper/strategic-IR connection), Delphi consensus and validity indicators (CVR calibration, Kendall's W), AHP aggregation and consistency (CR), and missing response processing. In terms of literature numbers, we have increased the number of methodology references in Methods to more than 15 recent studies.

5. Results and interpretation reviewer request complements

5-1. Request reviewer

• The resulting part presents only weights for industry-specific RSU governance standards, which are limited in interpretation and difficult for readers to understand industry-specific differences and implications

5-2. Author Supplements

• Enhanced interpretation of Delphi results. Beyond simply maintaining/deleting items and reporting statistics, we added a dedicated sub-clause (Section 4.4, "Comprehensive Interpretation of Delphi Results") that interprets mean, standard deviation (SD), CVR and Kendall's W by process stage/professional group. This explained where consensus is strongly formed and which items are contextually distributed and why.

• In case of Strengthen interpretation of AHP results by industry, AHP results were segmented by industry, adding and reinforcing the interpretation description by industry (Sections 5.2.1–5.2.4) by industry. By linking priority patterns in each industry with industrial structural characteristics such as market sensitivity, asset intensity and project cycle, talent market dynamics, and operational complexity, we complement the reader's understanding of "why a particular item is drawn to the top of the industry" beyond simple ranking.

• In case of Comprehensive comparison and implications between industries, in order to provide a quick glimpse of differences between industries, a new section on comparison and integration (Section 5.2.5) was established and the integrated priority table (Table 23) was presented. In addition, the commonalities and differences of each industry group and the implications based on them were discussed in an integrated manner.

• Augmented the supporting literature. To support the newly added interpretation, 26 new citations were added to Results/Discussion (New Insertion Reference Number: [135]–[160] based on the revised manuscript). These citations are reinforced around recent empirical and meta-analysis studies related to incentive design, performance management, industry heterogeneity, ESG/energy transformation, audit and internal control, and project/core talent-driven industry context. With the above modifications, this paper clearly explains why priorities vary by industry, not just by industry weight presentation, and improves the results to lead to actionable RSU governance design implications.

---

## [Decision Letter · Decision Letter 1]

8 Feb 2026

Strategic Equity Compensation A Delphi-AHP Approach to Industry-Specific RSUs Governance Design

PONE-D-25-36569R1

Dear Dr. Park,

We’re pleased to inform you that your manuscript has been judged scientifically suitable for publication and will be formally accepted for publication once it meets all outstanding technical requirements.

Kind regards,

Dipendra Karki, Ph.D.

Academic Editor

PLOS One

Additional Editor Comments (optional):

Reviewers' comments:

Reviewer's Responses to Questions

**Comments to the Author**

Reviewer #1: All comments have been addressed

Reviewer #2: All comments have been addressed

Reviewer #3: All comments have been addressed

2. Is the manuscript technically sound, and do the data support the conclusions?

Reviewer #1: Yes

Reviewer #2: Yes

Reviewer #3: Yes

3. Has the statistical analysis been performed appropriately and rigorously?

Reviewer #1: I Don't Know

Reviewer #2: Yes

Reviewer #3: Yes

4. Have the authors made all data underlying the findings in their manuscript fully available?

Reviewer #1: Yes

Reviewer #2: Yes

Reviewer #3: Yes

5. Is the manuscript presented in an intelligible fashion and written in standard English?

Reviewer #1: Yes

Reviewer #2: Yes

Reviewer #3: Yes

Reviewer #1: I would add the term Restricted Stock Units (RSUs) written out in full, considering that it only appears in the Introduction. Explaining the term right at the beginning can make it easier for the reader to understand, as well as aid in searches.

Review acronyms and write them out in full when possible, at least the first time they are mentioned in the text.

I would add a brief description of the organization of Figures 1 and 2, as well as lines 470-471 and 577-578 describing the figures in more detail, to do justice to their construction and integration into the article's text.

Reviewer #2: The paper addresses an interesting topic and has the potential to make a meaningful contribution to the field.

Reviewer #3: General Comments

The authors have been highly responsive to the previous round of reviews. The manuscript has evolved from a largely descriptive study into a theoretically grounded research paper. The inclusion of Behavioral Finance and Governance Reform frameworks provides the necessary academic context that was missing in the initial submission. I recommend tightening the Introduction to focus more sharply on the core research gap and the study’s specific objectives. A more streamlined opening would allow readers to reach the problem statement and methodology with greater clarity.

Specific Points

Methodological Rigor:

The clarification of the three-round Delphi process is an important correction. The use of the Content Validity Ratio (CVR) and Kendall’s W provides the level of statistical confidence required for a study of this nature. The distinction between the “Gatekeeper” expert group (Legal/Accounting) and the “Execution” group (HR/Strategy) is particularly insightful.

Literature Review:

The addition of 26 citations—particularly those addressing the structural limitations of stock options relative to the institutional advantages of RSUs—substantially strengthens the problem framing in the Introduction.

Industry-Specific Insights:

Section 5.2.5 and Table 23 are valuable additions. By contrasting the emphasis on “Talent Retention” in the technology sector with the focus on “Operational Complexity” in heavy industry, the paper provides actionable insights that clearly justify its focus on industry heterogeneity.

Discussion and Interpretation:

The new sub-clauses in Section 4.4 and the expanded discussion in Section 5 effectively move the analysis beyond what the data show to why these patterns emerge.

Major Suggestions for Final Polish

Tighten the Introduction:

Consider removing overly detailed empirical discussions to ensure a more direct path to the problem statement and research objectives.

Reference Placement and Consistency:

The addition of 26 new citations significantly enriches the manuscript’s scholarly depth. However, I encourage the authors to “front-load” these references by integrating them into the Literature Review (Section 2), rather than introducing them for the first time in the final discussion.

Additionally, please conduct a final reference check to ensure that all in-text citations—particularly the newly added references [135]–[160]—are correctly aligned with the bibliography.

Establishing the theoretical framework earlier will allow the Results section (Section 4) to remain a focused, data-driven presentation of findings, free from additional interpretative citations.

Geographical Scope:

While the paper is firmly grounded in the Korean corporate context, a brief concluding remark on how the findings might translate to other civil-law versus common-law jurisdictions would enhance the manuscript’s international relevance.

Attrition Reporting:

Please ensure that the manuscript explicitly states whether any experts dropped out between Round 1 and subsequent rounds of the Delphi survey.

**Do you want your identity to be public for this peer review?** For information about this choice, including consent withdrawal, please see our Privacy Policy

Reviewer #1: No

Reviewer #2: No

Reviewer #3: No

---

## [Editor Report · Acceptance letter]

PONE-D-25-36569R1

PLOS One

Dear Dr. Park,

I'm pleased to inform you that your manuscript has been deemed suitable for publication in PLOS One. Congratulations! Your manuscript is now being handed over to our production team.

Kind regards,

on behalf of

Dr. Dipendra Karki

Academic Editor

PLOS One